



# Description and application of a distributed hydrological model based on soil–gravel structure in the Qinghai–Tibet Plateau

Pengxiang Wang[1,2], Zuhao Zhou[1*], Jiajia Liu[1], Chongyu Xu[2,6], Kang Wang[2], Yangli Liu[3], Jia Li[4], Yuqing Li[5], Yangwen Jia[1], Hao Wang[1]

[1]State Key Laboratory of Simulation and Regulation of Water Cycle in River Basin, China Institute of Water Resources and Hydropower Research, Beijing 100038, China
[2]State Key Laboratory of Water Resources and Hydropower Engineering Science, Wuhan University, Wuhan 430072, China
[3]China Power Construction Group Guiyang Engineering Corporation Limited, Guiyang 550081, China
[4]Bureau of South to North Water Transfer of Planning, Designing and Management, Ministry of Water Resources, Beijing 100038, China
[5]Department of Water Resources and Civil Engineering, Tibet Agriculture and Animal Husbandry College, Nyingchi 860000, China
[6]Department of Geosciences, University of Oslo, Oslo, Norway

*Correspondence to*: Zuhao Zhou (zhzh@iwhr.com)

**Abstract.** The Qinghai–Tibet Plateau, known as the "Asian Water Tower", has a thin soil layer with a thick gravel layer underneath. Its unique geological structure, combined with widespread snow and frozen soil in this area, profoundly affect the water circulation processes of the entire region. To thoroughly study the water cycle mechanism of the Qinghai–Tibet Plateau, this study considered the geological and climatic characteristics of this area and selected the Niyang River Basin as the study area. The Water and Energy transfer Processes in the Qinghai–Tibet Plateau (WEP-QTP) model was constructed based on the original Water and Energy transfer Processes in Cold Regions (WEP-COR) model. This model divides the single soil structure into two types of media: the soil layer and gravel layer. In the non-freeze–thaw period, two infiltration models based on the dualistic soil–gravel structure were developed based on the Richards equation in non-heavy rain periods and the multi-layer Green–Ampt model in heavy rain periods. During the freeze–thaw period, a hydrothermal coupling model based on the continuum of the snow–soil–gravel layer was constructed. This distributed hydrological model can dynamically simulate the changes in frozen soil and flow processes in this area. The addition of the gravel layer corrected the original model's overestimation of the moisture content of the soil layer below the surface soil and reduced the moisture content relative error (RE) from 33.74 % to −12.11 %. The addition of the snow layer not only reduces the temperature fluctuation of the surface soil, but also works with the gravel layers to revise the original model's overestimation of the freeze–thaw speed of the frozen soil. The temperature RE was reduced from −3.60 % to 0.08 %. In the non-freeze–thaw period, the dualistic soil–gravel structure improved the regulation effect of groundwater on flow, stabilizing the flow process. The maximum RE at the flow peak and valley decreased by 88.2 % and 21.3 %, respectively. In the freeze–thaw period, by considering the effect of the snow–soil–gravel layer continuum, the change in the frozen soil depth of WEP-QTP lags behind that of WEP-COR by approximately one month. There was more time for the river groundwater recharge, which better shows the "tailing" process after October. The flow simulated by the WEP-QTP model was more accurate and closer to the actual measurements, with





Nash > 0.75 and |RE| < 10 %. The improved model reflects the effects of the Qinghai-Tibet Plateau special environment on the hydrothermal transport and water cycle process and is suitable for hydrological simulation of the Qinghai-Tibet Plateau.

## 1 Introduction

The Qinghai–Tibet Plateau, known as the "Asian Water Tower", is a typical cold mountainous area with low latitude and high
altitude. The region has a unique geology and landform, is sensitive to climate change (Liu et al., 2019), and plays an important role in ensuring the security of water resources in China and Southeast Asia. The impact of the extensive glacier, snow cover, and permanent and seasonal frozen soil on the water circulation processes of the entire area cannot be ignored.

Frozen soil plays an important role in the hydrological processes of cold regions (Chen et al., 2014; Kurylyk et al., 2014). During the process of soil freezing, ice blocks the majority of the pores in the soil, hinders infiltration, and affects water
movement in the soil. With regard to the hydrological cycle, the frozen soil layer prevents rainfall and snowmelt infiltration, forcing surface runoff down the slope, which may lead to severe flash floods. Conversely, it also affects the quantity of groundwater recharge supplemented by infiltration and the distribution ratio of surface runoff between rivers and lakes (Ireson et al., 2013; Larsbo et al., 2019). Studies have shown that the degradation of permafrost is one of the main causes of reductions in both the groundwater table and lake water level, as well as swamp and grassland degradation in the source regions of the
Yangtze and Yellow Rivers (Cheng and Wu, 2007). In addition, snow is also a factor that cannot be ignored in the hydrological cycle in cold regions. One notable point is that precipitation is stored in the form of snow in winter and melts quickly after the temperature rises, which may even cause spring floods. Another notable point is that the accumulation and redistribution of snow also affect the temporal and spatial distribution of water resources (Dutra et al., 2012).

In the context of climate change, it is necessary to study the influence of the above factors on the hydrological cycle.
Hydrological models in cold regions have made some progress in this respect, such as the SHAWDHM model (Zhang et al., 2013), GEOtop model (Rigon et al., 2006), Cold Regions Hydrological Model (CRHM) (Pomeroy et al., 2007), and Variable Infiltration Capacity (VIC) model (Cherkauer and Lettenmaier, 2003), which can simulate the water cycle processes in general cold regions including snow and frozen soil to a certain extent. Nonetheless, soil water and heat transfer are relatively complex processes that are influenced by many factors (Watanabe and Kugisaki, 2017) such as soil structure (Dai et al., 2019;
Franzluebbers, 2002) and temperature conduction under snow cover (Lundberg et al., 2016). In the Qinghai–Tibet Plateau, although the water cycle processes are similar to those in a general cold region, there are also particular differences that cannot be ignored.

During the geological formation process, the Indian plate and the Eurasian plate continuously collided and the crustal movement in the Qinghai–Tibet Plateau was active, so gravel and rock debris are common in the soil (Arocena et al., 2012).
In addition, under a strong freeze–thaw cycle, the accumulation of humus and the decomposition and leaching of minerals are





weak, soil formation is slow, and the soil layer is thin (Deng et al., 2019). Therefore, the geological structure of the Qinghai–Tibet Plateau is special, with a thick gravel layer under the thin soil layer.

Soil mixed with gravel has different hydraulic and thermodynamic properties compared with soil alone, and the degree of difference is affected by the sand and gravel content (Zhang et al., 2011). When the sand and gravel content is low, the gravel
changes the soil structure and increases the distance of soil water movement; the saturated hydraulic conductivity of soil decreases with an increase in the gravel content (Childs and Flint, 1990; Mehuys et al., 1975). However, when the sand and gravel content exceeds a certain level, connected macropores are formed in the soil, and the soil's saturated hydraulic conductivity increases along with the content (Beibei et al., 2009). In the heat transfer process, the greater thermal conductivity and heat capacity of gravel compared with those of dry soil affect the geothermal flux (Yi et al., 2013). The dualistic soil–
gravel structure changes the regional water cycle processes by affecting deep leakage and hydrothermal conduction.

Therefore, it is of great significance to consider the influence of the dualistic soil–gravel structure on the hydrothermal coupling and flow simulation of the hydrological model when simulating the hydrology in the Qinghai–Tibet Plateau. The purpose of this study was to: (1) develop infiltration models based on the dualistic soil–gravel structure in non-heavy and heavy rain periods during the non-freeze–thaw period, (2) develop a hydrothermal coupling method based on the continuum of snow–
soil–gravel layer through field water and heat monitoring experiments during the freeze–thaw period, and (3) study the effect of the dualistic soil–gravel structure on the hydrological cycle by building the distributed water cycle model (WEP-QTP) for the Niyang River Basin, a tributary of the Yarlung Zangbo River in the Qinghai–Tibet Plateau.

## 2 Materials and methods

### 2.1 Study sites and data

#### 2.1.1 Study area

The Niyang River is located on the left bank of the lower reaches of the Yarlung Zangbo River, between 29°28'–30°38' N and 92°10'–94°35' E in the Linzhi area of southeastern Tibet. It originates from Cuomoliang Mountain on the west side of the Mila Mountain in the Tibet Autonomous Region of China at an altitude of approximately 5000 m above sea level (a.s.l.). The Niyang River flows through Gongbujiangda County and Bayi Town from west to east and finally flows into the Yarlung Zangbo River
in the Bayi District of Nyingchi City, with a drop of 2080 m and an average slope drop of 0.73 %. The basin is approximately 230 km long from east to west and 110 km wide from north to south. The area of the watershed is 17 535 km², ranking fourth among the five tributaries of the Yarlung Zangbo River, and its runoff is second only to that of Palungzangbu. The Niyang River Basin is located at the intersection of Tibet's east–west and north–south mountain ranges. The terrain in the watershed is complex, with staggered large and small mountains and large elevation fluctuations. The elevation of the river valley is
generally 3000–4000 m a.s.l. The elevation of most mountain peaks on both sides of the valley is approximately 5000 m a.s.l, reaching up to 6870 m a.s.l. The Niyang River Basin belongs to the plateau temperate monsoon climate zone. The multi-year



average precipitation is affected by the Indian Ocean tropical ocean monsoon. Under the effect of the Indian low pressure, the southwest monsoon pushes a large amount of warm and humid air from the Bay of Bengal along the Yarlung Zangbo River Valley to the Niyang River Basin, causing precipitation in the basin with heavy rainfall and large vertical changes. The average

annual precipitation in the basin is 1416 mm, and the average annual temperature is approximately 8 °C. Obvious temperature changes occur from east to west with elevation.

In this study, a monitoring experiment of the coupling processes of water and heat during the seasonal freezing and thawing period was carried out on the mountainside of the Sejila Mountain in the lower reaches of the Niyang River Basin. The longitude and latitude of the study site are 94°21′45″ E and 29°27′12″ N, respectively, and the altitude is 4607 m a.s.l. The

experimental period was 2016–2017, and the freeze–thaw period was from November 2016 to March 2017. The basic situation of the watershed and the location of the experimental points are shown in Fig. 1.

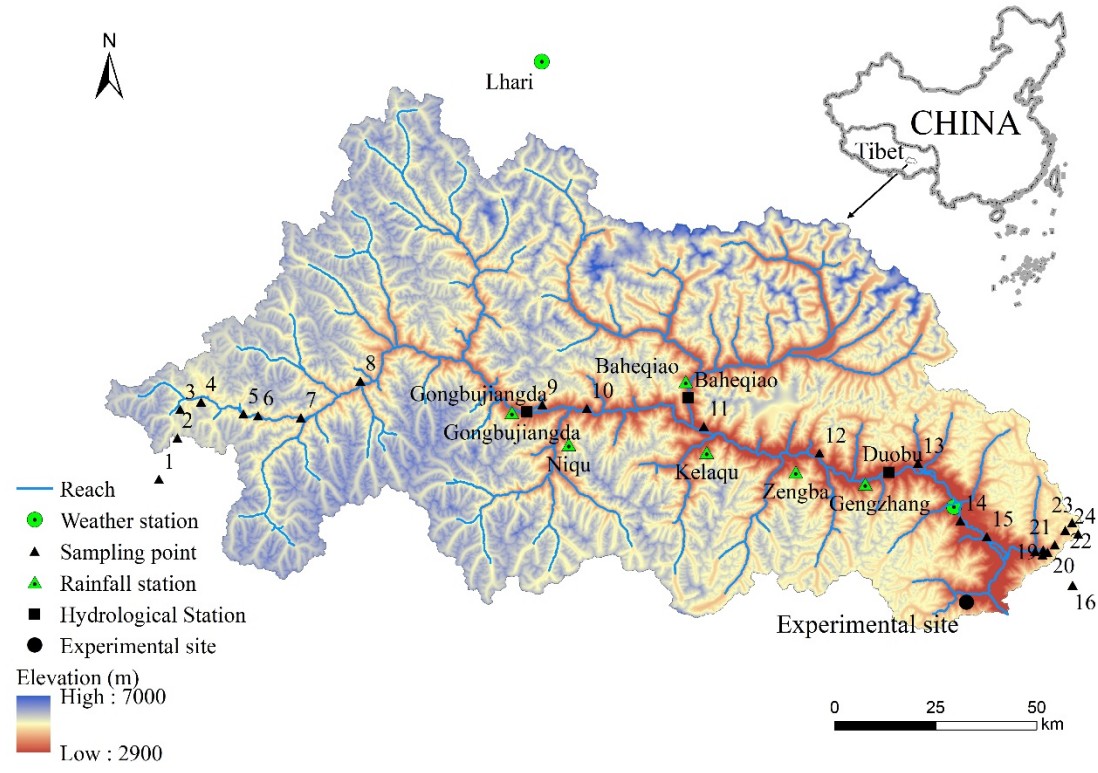

**Figure 1: Basic situation and station distribution of the Niyang River Basin**


### 2.1.2 Data description

The data required for this study were mainly divided into two categories: the first category was the data required for model construction (mainly including meteorology, geology and landform, terrain, soil type, land-use type, vegetation index, and





glacier data), and the second category was the data used to verify the model results (including historical and experimental
monitoring data).

Temperature, relative humidity, sunshine hours, and wind speed data were collected by the Nyingchi Meteorological Station
in the basin and the Jiali Meteorological Station outside the basin, from 1961–2018. The data were obtained from the China
Meteorological Data Network (http://data.cma.cn). In addition to the two meteorological stations in Nyingchi and Jiali, the
rainfall data sources also included six rainfall stations (2013–2015) including Gongbujiangda, Gengzhang, Baheqiao, Niqu,
Kelaqu, and Zengba in the watershed and the contour map of annual precipitation in the Tibet Water Resources Bulletin (2012–
2017). The temperature, relative humidity, sunshine hours and wind speed in the basin were interpolated from the
meteorological station data by the reversed distance squared method with elevation correction. As for the precipitation data,
the rainfall stations in the study area were concentrated in the valley (Fig. 1). If only these data were used to interpolate
precipitation, there would be large errors in high altitude areas, affecting the accuracy of runoff simulation. Therefore, in this
study, the precipitation–elevation relationship was first determined based on the contour map of annual precipitation in the
Tibet Water Resources Bulletin and the station precipitation data. Then, the daily precipitation data in the basin were obtained
through elevation interpolation (Wang et al., 2017).

Due to the combined effects of plate tectonics, weathering, and erosion, a unique geological structure was formed in the
Qinghai–Tibet Plateau with a thin soil layer on the top and a thick gravel layer on the bottom (Fig. 2). According to the
geological characteristics of the Qinghai–Tibet Plateau, the Niyang River Basin, a first-level tributary of the Yarlung Zangbo
River, was selected in this study to represent a typical area. From the source of the river to the estuary, 24 sampling points at
different altitudes were selected to conduct field investigations on the soil texture (Fig. 1). Among them, points 1 to 16 were
along the river, and points 17 to 24 were from the foot of the mountain to the peak.

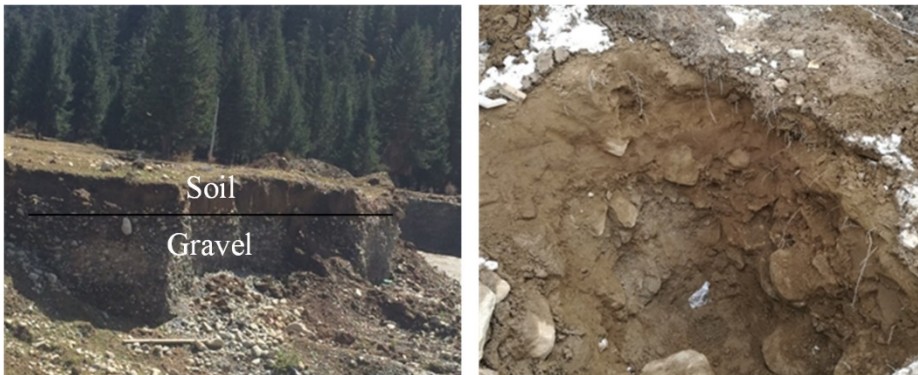


**Figure 2: Soil and gravel structure of the Niyang River Basin**

The soil thicknesses and compositions at the 24 sampling points were measured and analyzed. The soil layer of the Niyang
River Basin is mainly sandy loam, with average sand, silt, and clay contents of 55.89 %, 31.2 %, and 12.91 %, respectively.





The gravel layer is mainly round gravel, containing pebbles; the gravel content is approximately 50 %−65 %, the clay content is 5 %−10 %, and the pores are filled with medium and fine-grained sand. The thickness of the soil layer gradually decreases from the foot to the peak of the mountain. It is approximately 40 cm on the hillside with higher altitude and increases to more than 100 cm in the valley.

The elevation data (Digital Elevation Model, DEM) used in this study were from the SRTM90 (Shuttle Radar Topography

Mission), which is jointly measured by the National Aeronautics and Space Administration (NASA) and the National Imagery and Mapping Agency (NIMA) with an accuracy of 90 m.

Soil type data were obtained from the second national soil census and "Chinese Soil Records". Land-use data came from the Resource Environment Science and Database Center, Institute of Geographic Sciences and Natural Resources Research, Chinese Academy of Sciences (http://www.resdc.cn), and the data resolution was 30 m.

Moderate-resolution Imaging Spectroradiometer (MODIS) data from 2000–2017 were selected as the data source. Among them, the leaf area index accuracy was 500 m, and the normalized difference vegetation index accuracy was 250 m; these were mainly used to calculate evaporation and vegetation interception processes, respectively.

The glacier data included China's second glacier inventory data set (1:100 000) and Landsat TM/ETM+/OLI remote sensing images. The second glacier inventory data comes from the China Cold and Arid Regions Science Data Center

(http://westdc.westgis.ac.cn/). Landsat data came from the data sharing platform of the United States Geological Survey (USGS) (http://glovis.usgs.gov/). ENVI software was used to extract glaciers, and the boundaries of the glaciers were finally determined with reference to Google Earth imagery. According to China's second glacier cataloging rules, the glaciers in the basin were classified and the glacier area was calculated. The volume of the glacier was calculated by the area-volume empirical formula (Grinsted, 2013; Radić and Hock, 2010).

Model verification data included historical data and experimental monitoring data. Historical data included daily measured flow data from the Gongbujiangda Hydrological Station (2013–2016, 2018), the Baheqiao Hydrological Station (2013–2014), and the Duobu station (2013–2018). The experimental monitoring data included the soil temperature and volumetric water content of the experimental site from 2016–2017. At the test point, a time-domain reflectometry sensor for monitoring the water content of the liquid water, a PT100 sensor for measuring the temperature, and a TensionMark sensor for measuring the

potential of the substrate were installed every 10 cm in the vertical depth of the experimental pit at a depth of 1.6 m. Water, heat, and potential energy were automatically monitored during freezing and thawing.

## 2.2 Model and theory

### 2.2.1 Introduction of WEP-COR

The WEP-QTP model was developed based on the Water and Energy transfer Processes in Cold Regions (WEP-COR) model.

For the sake of understanding and comparison, the WEP-COR model is briefly introduced in this section. The vertical structure of WEP-COR is divided into the vegetation canopy or building interception layer, the surface depression storage layer, the





aeration layer, the transition zone layer, and the groundwater layer. To accurately simulate the changes in soil moisture and heat from the surface to the deep layers and to reflect the influence of soil depth on the evaporation of bare soil and the water absorption and transpiration of vegetation roots, the aerated zone soil is divided into 11 layers (Fig. 3a). Among them, $R_{surface}$

represents runoff from the surface, and $R_i$ represents lateral flow or soil flow in the $i$-th layer of soil. $R_i$ is related to slope and soil moisture content. $E_1$ represents soil evaporation, $E_r$ represents vegetation transpiration, $Q_i$ is gravity drainage of the $i$-th layer, $P$ is precipitation, $T_a$ is atmospheric temperature, $T_i$ is the temperature of the $i$-th layer, and $G_i$ is the heat flux caused by the temperature difference between the $i$-th layer of soil and the adjacent soil layers. The thickness of the first and second layers was set to 10 cm, and the thickness of layers 3–11 was set to 20 cm.


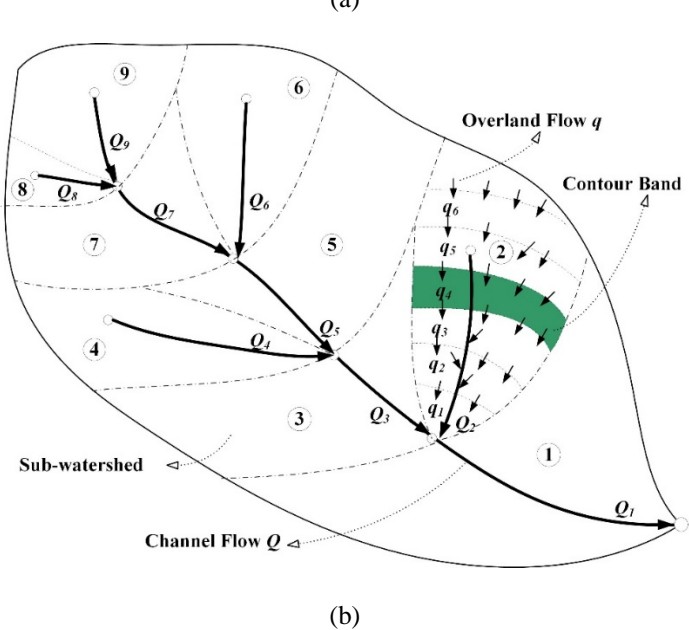

(a)

(b)

**Figure 3: Vertical (a) and horizontal (b) structure of the WEP-COR model**


The WEP-COR model divides the infiltration process into two scenarios for simulation: the heavy rain period and non-heavy rain period. The division standard is whether the daily rainfall exceeds 20 mm. In the non-heavy rain period, the Richards equation is used for daily scale simulation (Jia et al., 2009). During the heavy rain period, the multi-layer unsteady rainfall

Green–Ampt model proposed by Jia and Tamai is used (Jia and Tamai, 1998). The relationship between the water and heat transport of frozen soil is mainly manifested in the dynamic balance of the moisture content of the unfrozen water and the negative temperature of the soil. According to the principle of energy balance, the energy change of each layer in the freeze–thaw system was used for the soil temperature change and water phase change in the system.

In terms of the horizontal structure, the WEP-COR uses the contour bands inside small sub-basins as the basic calculation unit

(Fig. 3b), and it can fully consider the vertical changes of vegetation, soil, air temperature, precipitation, and other factors in the basin with the elevation. Each unit is divided into five types according to the land-use type: water body, soil–vegetation, irrigated farmland, non-irrigated farmland, and impervious area. The calculation result of the water and heat flux in each type was weighted by area to obtain the water and heat flux of the contour band. Evapotranspiration of water and soil was calculated using the Penman formula, and the vegetation canopy evaporation was calculated using the Penman–Monteith formula

(Monteith, 1973). The subsurface runoff was calculated based on slope and soil hydraulic conductivity, and groundwater movement was calculated by the Boussinesq equation (Zaradny, 1993). The confluence of overland flow and channel flow were calculated using the kinematic wave method (Jia et al., 2001). The "degree-day factor method" (Hock, 1999) was used to calculate the quantity of glacier melting, and the runoff from the melting of glaciers was directly added to the corresponding hydrological calculation unit. For other details of the WEP-COR model, please refer to Li et al. (2019).

**2.2.2 Model improvement**

Based on the WEP-COR model, this study developed the improved WEP-QTP (WEP-Qinghai–Tibet Plateau) model. In contrast to the general cold areas where the WEP-COR model is applied, the widespread dualistic soil–gravel structure in the Qinghai–Tibet Plateau has a great impact on the water cycle processes in the basin. According to the geological characteristics of the Qinghai–Tibet Plateau, this study improved the simulation methods of the non-freeze–thaw period and the freeze–thaw

period.

In the non-freeze–thaw period, the calculation object of water movement above the groundwater was defined as the dualistic soil–gravel structure (Fig. 4a). The upper layer is soil, and its thickness and number of layers are determined by the location of the calculation unit; the thickness of the soil layer gradually decreases from the foot to the peak of the mountain. The lower layer is the gravel layer (mixed layer of soil and gravel).



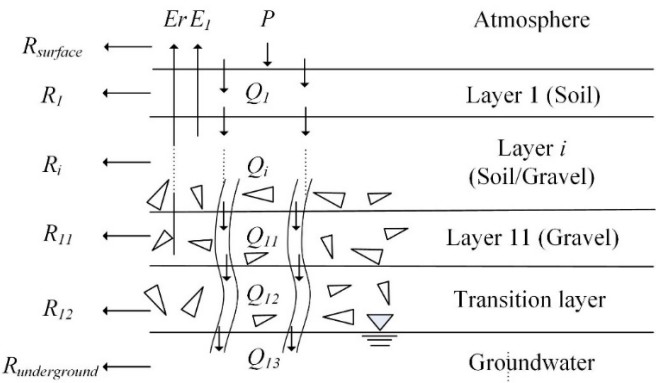

(a)

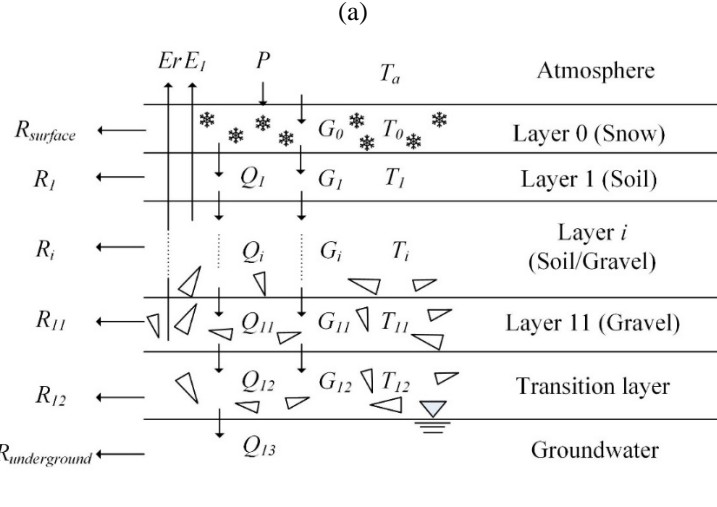

(b)

**Figure 4: Layered calculation structure of the dualistic "soil–gravel" structure (a) and the "snow–soil–gravel layer" continuum (b)**

In non-heavy rain periods, as in the WEP-COR model, the water movement process was described by the one-dimensional vertical Richards equation as follows:

$$\frac{\partial \theta_l}{\partial t} = \frac{\partial}{\partial z}\left[D(\theta_l)\frac{\partial \theta_l}{\partial z}\right] - \frac{\partial K(\theta_l)}{\partial z} \tag{1}$$

where $\theta_l$ is the volumetric content of liquid water in the soil or gravel layer (cm³/cm³); $D(\theta_l)$ and $K(\theta_l)$ are the unsaturated soil hydraulic diffusivity (cm²/s) and hydraulic conductivity (cm/s); and $t$ and $z$ are the time and space coordinates (positive vertically downward).

The Van Genuchten function (Van Genuchten, 1980) was used to describe the upper soil water retention curves:

$$\frac{\theta_l - \theta_r}{\theta_s - \theta_r} = \frac{1}{[1+(\alpha h)^n]^m} \tag{2}$$





where $\theta_s$ is the saturated water content (cm³/cm³); $\theta_r$ is the residual water content (cm³/cm³); $h$ is the matric suction (cm); α is an empirical parameter (cm⁻¹); $n$ and $m$ are empirical parameters affecting the shape of the retention curve; and $m = 1–1/n$. However, because gravel can neither conduct nor store water, the gravel, which accounts for 50 %–65 % of the gravel layer, hinders the movement of water and affects the water retention curves (Cousin et al., 2003). Therefore, the revised formula for water retention properties of the soil–gravel mixture was used to describe the lower gravel layer water retention curves (Wang

et al., 2013):

$$\frac{\theta_l-\theta_r}{\theta_s-\theta_r} = Ah^{-\lambda}\left(1 - \omega_{gravel}\right) \tag{3}$$

where $A$ is an empirical parameter, $\lambda$ is the pore-size distribution parameter ($\lambda < 1$), and $\omega_{gravel}$ is the volume ratio of the gravel in the gravel layer.

In the heavy rain period, the multi-layer Green–Ampt equation was used to calculate the infiltration process when the

infiltration front (INF) was in the soil layer (Fig. 5), which is the same as in WEP-COR. When the INF reached the $m$-th layer of soil, the soil infiltration capacity was calculated by the following formulas:

$$f = k_m(1 + \frac{A_{m-1}}{B_{m-1}+F}) \tag{4}$$

$$A_{m-1} = (\sum_1^{m-1} L_i - \sum_1^{m-1} \frac{L_i k_m}{k_i} + SW_m)\Delta\theta_m \tag{5}$$

$$B_{m-1} = \left(\sum_1^{m-1} \frac{L_i k_m}{k_i}\right)\Delta\theta_m - \sum_1^{m-1} L_i\Delta\theta_i \tag{6}$$

where $f$ is the infiltration capacity (mm/h); $A_{i-1}$ is the total water capacity of the soil above the $i$ layer (mm); $B_{i-1}$ is the error caused by the different soil moisture content of the soil above the $i$ layer (mm); $F$ is the cumulative infiltration (mm); $k_i$ is the hydraulic conductivity of the $i$-th soil layer (mm/h); $L_i$ is the soil thickness of the $i$-th layer (mm); $SW_m$ is the capillary suction pressure at the INF of the $m$-th layer (mm); and $\Delta\theta_i = \theta_s - \theta_l$.

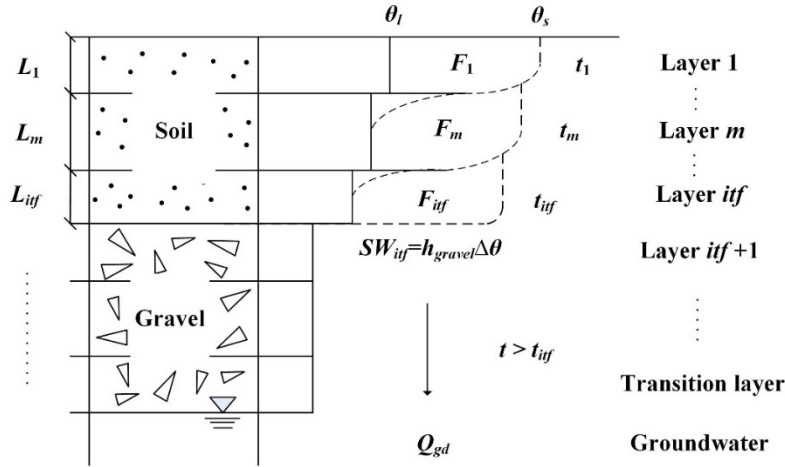


**Figure 5: Cumulative infiltration process of the WEP-QTP model**





The cumulative infiltration quantity $F$ when the INF reaches the $m$-th layer is calculated based on whether there is water accumulation on the ground surface. If the ground surface has accumulated water when the INF reaches the $m$–1th layer, Eq. (7) was used; otherwise, Eq. (8) was used (Jia and Tamai, 1998):

$$F - F_{m-1} = k_m(t - t_{m-1}) + A_{m-1}\ln(\frac{A_{m-1}+B_{m-1}+F}{A_{m-1}+B_{m-1}+F_{m-1}}) \tag{7}$$

$$F - F_p = k_m(t - t_p) + A_{m-1}\ln(\frac{A_{m-1}+B_{m-1}+F}{A_{m-1}+B_{m-1}+F_p}) \tag{8}$$

$$F_{m-1} = \sum_1^{m-1} L_i\Delta\theta_i \tag{9}$$

$$F_p = A_{m-1}(\frac{I_p}{k_m} - 1) - B_{m-1} \tag{10}$$

$$t_p = t_{m-1} + (F_p - F_{n-1})/I_p \tag{11}$$

where $t$ is the time; $t_{m-1}$ is the time when the INF reaches the interface between the $m-1$ and $m$ layers; $t_p$ is the start time of the water accumulation; $F_p$ is the cumulative infiltration quantity at $t_p$; and $I_p$ is the precipitation intensity at $t_p$.

When the INF moves to the interface of the soil and gravel (Layer $itf$), the front movement slows down because the water suction of the gravel layer is less than that of the soil (Mao and Shang, 2010). Until the water has the same potential energy in the soil and the gravel, the INF breaks through the critical surface, and then the infiltration rate stabilizes (Fig. 5). Therefore, a new multi-layer Green–Ampt model based on the soil–gravel structure was proposed as follows:

$$f_{gravel} = k_{soil}(1 + \frac{A_{itf}}{B_{itf}+F_{itf}}) \tag{12}$$

where $f_{gravel}$ is the stable infiltration rate after breaking through Layer $itf$; $k_{soil}$ is the saturated hydraulic conductivity of the soil layer (mm/h); $A_{itf}$ is the total water capacity of the soil above the interface (mm); $B_{itf}$ is the error caused by the different soil moisture content of the soil above the interface (mm); and $F_{itf}$ is the cumulative infiltration when the front breaks through Layer $itf$ (mm).

The large portion of gravel in the gravel layer causes the formation of macropores, which are connected to form a fast channel for transporting water during heavy rains (Fig. 4a). After the INF breaks through the interface, the infiltration water preferentially recharges the groundwater through the macropores. At this time, the accumulated infiltration quantity is as follows:

$$F = F_{itf} + Q_{gd} \tag{13}$$

where $Q_{gd}$ is the quantity of groundwater recharge by infiltration, $Q_{gd} = f_{gravel}(t - t_{itf})$, and $t_{itf}$ is the time when the INF breaks through the interface.

During the freeze–thaw period, in addition to considering the impact of gravel on hydrothermal transfer, the higher reflectivity of snow to shortwave solar radiation and its contribution to heat insulation were also taken into consideration. The hydrothermal coupling simulation object was defined as the snow–soil–gravel layer continuum (Fig. 4b). A snow layer was added on top of the dualistic soil–gravel structure, the thickness of which was determined by the snow water equivalent and snow density.





For the heat transfer process, assuming that the upper boundary of the system is the atmosphere, which controls the input and
output of the system energy. When there is snow on the surface, the atmosphere first exchanges energy with the snow layer,
and then the snow layer exchanges energy with the soil. When there is no snow, the atmosphere directly exchanges energy
with the soil, and the upper boundary energy can be calculated by meteorological elements. The lower boundary at the bottom
is the transition layer or groundwater layer, assuming that it maintains a constant temperature.

The energy balance equation of the surface can be expressed by the following equation (Jia et al., 2001):

$$RN = LE + H + G \tag{14}$$

where $RN$ is the net radiation flux (MJ/m²/d); $LE$ is the latent heat flux (MJ/m²/d), which was calculated from the melting and
evaporation of the snow layer and the freezing, thawing, and evapotranspiration of the soil layer moisture; $H$ is the sensible
heat flux (MJ/m²/d), which was obtained from the remainder of the surface energy balance equation; and $G$ is the heat flux
(MJ/m²/d) conducted into the snow or soil, which was determined by the temperature difference between the soil or snow and
the atmosphere near the surface.

The surface snow or soil temperature (when there was no snow cover) was calculated by the forced recovery method (Douville
et al., 1995; Pitman et al., 1991). For soil and gravel layers, the average temperature was represented by the temperature in the
middle of the layer. The temperature difference between the atmosphere and the surface is the source of heat conduction; after
the surface temperature was determined, the heat flux and temperature of each layer were calculated by the following formula
(Shang et al., 1997; Wang et al., 2014):

$$C_v \frac{\partial T}{\partial t} = \frac{\partial}{\partial z}\left[\lambda \frac{\partial T}{\partial z}\right] + L_f \rho_I \frac{\partial \theta_I}{\partial t} \tag{15}$$

where $C_v$ and $\lambda$ are the volumetric heat capacity (J/[m³·°C]) and thermal conductivity (W/[m·°C]) of the soil or gravel layer,
respectively; $L_f$ is the latent heat of ice melting ($3.35 \times 10^6$ J/kg); $T$ is the temperature (°C) of the soil or gravel layer; $\rho_I$ is the
ice density (kg/m³); $\theta_I$ is the volumetric content of ice in the soil or gravel layer (cm³/cm³); and $z$ is the layer thickness (m).

The redistribution of snow on the Qinghai–Tibet Plateau is affected more by the elevation difference than by wind. The daily
variation of snow water equivalent was calculated as follows:

$$S = S_p - S_d - S_m \tag{16}$$

where $S$ is the daily variation of snow water equivalent (mm/d); $S_p$ is the snow water equivalent from precipitation (mm/d), $S_p$
is equal to the daily precipitation when the average temperature of the day was $T_a < 2$ °C, otherwise $S_p = 0$; $S_d$ is the snow water
equivalent variation due to snow sliding down (mm/d), when the difference in snow thickness between contour bands in the
same sub-basin exceeds the threshold, the snow slides downwards until the snow thickness is the same; $S_m$ is the quantity of
snow melting equivalent (mm/d), which was calculated by the degree-day factor method (Hock, 1999) as follows:

$$S_m = d_f(T_a - T_S) \tag{17}$$

where $d_f$ is the degree-day factor (mm/[°C·day], i.e., 4 mm/[°C·day] in this study); $T_S$ is the critical temperature of snow
melting (°C, i.e., −1 °C in this study), assuming snow melt starts when $T_a$ is > $T_S$.





During the freeze–thaw period, only liquid water migrates. The soil vertical heat flux transfer can be written as follows (Shang et al., 1997; Wang et al., 2014):

$$\frac{\partial \theta_l}{\partial t} = \frac{\partial}{\partial z}\left[D(\theta_l)\frac{\partial \theta_l}{\partial z} - K(\theta_l)\right] - \frac{\rho_I}{\rho_l}\frac{\partial \theta_I}{\partial t} \tag{18}$$

where $\rho_l$ is the water density (kg/m³).

Temperature is the driving force of the water phase change. The relationship between the water and heat transport of frozen soil is mainly manifested in the dynamic balance of the moisture content of the unfrozen water and the negative temperature of the soil:

$$\theta_l = \theta_m(T) \tag{19}$$

where $\theta_m(T)$ is the maximum unfrozen water moisture content corresponding to a negative soil temperature.

For the snow layer added to improve the model, the main hydrothermal parameters include thermal conductivity, volumetric heat capacity, and snow density. The calculation formulas of each parameter are as follows:

Snow density was calculated by Eq. (20) (Hedstrom and Pomeroy, 1998):

$$\rho_s = \begin{cases} 67.9 + 51.3e^{T_a/2.6} & T_a \le 0 \\ 119.2 + 20T_a & T_a > 0 \end{cases}. \tag{20}$$

The thermal conductivity and volumetric heat capacity of snow were calculated as follows (Goodrich, 1982; Ling and Zhang,

330    2006):

$$\lambda_s = \begin{cases} 0.138 - \frac{1.01\rho_s}{1000} + 3.233\left(\frac{\rho_s}{1000}\right)^2 & 156 < \rho_s \le 600 \\ 0.023 + \frac{0.234\rho_s}{1000} & \rho_s \le 156 \end{cases} \text{ and} \tag{21}$$

$$C_{Vs} = 2.09\rho_s \times 10^3, \tag{22}$$

where $\rho_s$ is the snow density (kg/m³); $T_a$ is the atmospheric temperature (°C); $\lambda_s$ is the thermal conductivity of the snow (W/[m·°C]); and $C_{Vs}$ is the volumetric heat capacity of the snow (J/[m³·°C]).

As opposed to a single soil medium, the presence of gravel has a great influence on the hydrothermal transfer parameters of the gravel layer. The main hydrothermal parameters of the soil–gravel layer include volumetric heat capacity, thermal conductivity, and soil hydraulic conductivity. The calculation formulas for each parameter were as follows:

Volumetric heat capacity were calculated by Eq. (23) (Chen et al., 2008):

$$C_V = (1 - \theta_s) \times C_s + \theta_l \times C_l + \theta_I \times C_I, \tag{23}$$

where $\theta_s$, $\theta_l$, and $\theta_I$ are the saturated volumetric water content, volumetric liquid water content, and volumetric ice content of the soil or gravel layer, respectively; $C_s$, $C_l$, and $C_I$ are the volumetric heat capacity (J/[m³·°C]) of the soil or gravel layer, water, and ice, respectively; at 0 °C, the soil and gravel layers have values of 1.93×10³ J/[m³·°C] and 3.1×10³ J/[m³·°C], respectively; and water and ice have values of 4.213×10³ J/[m³·°C] and 1.94×10³ J/[m³·°C], respectively.

The thermal conductivity calculation referred to the IBIS model, as follows (Foley et al., 1996):

$$\lambda = \lambda_{st} \times (56^{\theta_l} + 224^{\theta_I}), \text{ and} \tag{24}$$

$$\lambda_{st} = \omega_{gravel} \times 1.5 + \omega_{sand} \times 0.3 + \omega_{silt} \times 0.265 + \omega_{clay} \times 0.25, \tag{25}$$





where $\lambda$ and $\lambda_{st}$ are the actual thermal conductivity of the soil or gravel layer and the thermal conductivity in the dry state (W/[m·°C]), respectively, and $\omega_{gravel}$, $\omega_{sand}$, $\omega_{silt}$, and $\omega_{clay}$ are the volume ratios of the gravel, sand, silt, and clay, respectively. The hydraulic conductivity was calculated as follows (Chen et al., 2008):

$$K(\theta_l) = \begin{cases} K_S & \theta_l = \theta_s \\ K_S \left(\frac{\theta_l - \theta_r}{\theta_s - \theta_r}\right)^n & \theta_l \neq \theta_s \end{cases}, \qquad (26)$$

where $\theta_r$ is the residual water content of the soil or gravel layer; $K(\theta_l)$ is the hydraulic conductivity (cm/s) of the soil or gravel layer when the liquid water content is $\theta_l$; $K_s$ is the saturated hydraulic conductivity of the soil temperature correction (cm/s); and $n$ is Mualem's constant.

$K_s$ can be calculated as follows (Chen et al., 2008; Jansson, 2004):

$$K_s = \begin{cases} K & T > 0 \\ K(0.54 + 0.023T) & T_f \leq T \leq 0, \\ K_0 & T < T_f \end{cases} \qquad (27)$$

where $K$ is the initial saturated hydraulic conductivity (cm/s) and $K_0$ is the minimum hydraulic conductivity (cm/s) under freezing conditions. Considering the difference in the hydrodynamic properties of the soil and gravel layer, the $K_0$ value for soil was considered to be 0 cm/s. For the gravel layer, due to the larger pores, $K_0$ has a value of $> 0$ cm/s; $T$ is the temperature of the soil or gravel layer (°C); and $T_f$ is the critical temperature (°C) corresponding to the minimum hydraulic conductivity.

**2.3 Model evaluation criteria**

Data from January 2013 to December 2018 were used to evaluate the simulation results of daily flow rates at Gongbujiangda, Baheqiao, and Duobu stations. The performance of the model was first evaluated using a qualitative assessment via graphs and then assessed quantitatively using statistical metrics including the Nash–Sutcliffe efficiency (NSE) and relative error (RE). The NSE and RE were calculated as follows:

$$NSE = 1 - \frac{\sum_{i=1}^{N}(O_i - S_i)^2}{\sum_{i=1}^{N}(O_i - \overline{O_i})^2} \qquad (28)$$

$$RE = \frac{\sum_{i=1}^{N} S_i - \sum_{i=1}^{N} O_i}{\sum_{i=1}^{N} O_i} \times 100 \% \qquad (29)$$

where $N$ is the number of observations; $O_i$ is the observed value; $\overline{O}$ is the mean observed value; and $S_i$ is the simulated value.

**3 Results and discussion**

**3.1 Model calibration and validation**

We calibrated and verified the daily flow process of Gongbujiangda, Baheqiao, and Duobu stations—located upstream, on the largest tributary, and downstream, respectively—from 2013 to 2018. The data from Duobu station were split into two parts: data from 2013 to 2015 were used for calibration and those from 2016 to 2018 for validation. The discontinuous, measured flow data from Gongbujiangda and Baheqiao stations from 2013 to 2018 were used to verify the model.





The parameters of the model were mainly divided into four categories: underlying surface parameters, vegetation parameters,
soil parameters and aquifer parameters. All parameters have physical meaning and can be estimated based on observational
experimental data or remote sensing data. The sensitivity of the above four types of parameters was analyzed (Jia et al., 2006),
and the sensitivity of these parameters was divided into three levels: high, medium, and low. Highly sensitive parameters
included soil thickness, soil saturated hydraulic conductivity, and riverbed material permeability coefficient. The model was
calibrated according to the runoff process. The saturated hydraulic conductivity of the soil layer was 0.648 m/d, that of the
gravel layer was 4.32 m/d, and the riverbed conductivity was approximately 5.184 m/d. The thickness of the soil layer at the
mountaintop, mountainside, and foot of the mountain was 0.4 m, 0.6 m, and 1.0 m, respectively.

Figure 6 and Table 1 present the results of both calibration and verification periods of the daily flow data from Duobu station
and only the latter from Gongbujiangda and Baheqiao stations. The simulation results of the WEP-QTP model from the three
stations are consistent with the measured flow data. During the verification period, compared with the WEP-COR model, the
NSE of WEP-QTP increased and the RE decreased, thereby considerably improving the simulation effect of the model. The
water storage capacity and permeability of the aquifer were considerably improved owing to the addition of the gravel structure.
The WEP-QTP simulation flow process was smoother, and a large flow peak was not easily formed. In general, the WEP-QTP
model delivered an acceptable performance for the Niyang River Basin and achieved efficiency coefficients of NSE > 0.75
and RE < 10 % for the validation period. The simulated flow was able to be used for further analysis.

**Table 1: Model validation results for Gongbujiangda, Baheqiao, and Duobu stations**

| Model | Duobu | | | | Gongbujiangda | | Baheqiao | |
| | Calibration | | Validation | | Validation | | Validation | |
| | NSE | RE | NSE | RE | NSE | RE | NSE | RE |
|---|---|---|---|---|---|---|---|---|
| WEP-QTP | 0.89 | −5.8 % | 0.76 | 3.4 % | 0.79 | 0.01 % | 0.75 | −5.47% |
| WEP-COR | 0.69 | −4.65% | 0.31 | 0.01 % | 0.67 | 1.66 % | 0.40 | −2.38 % |





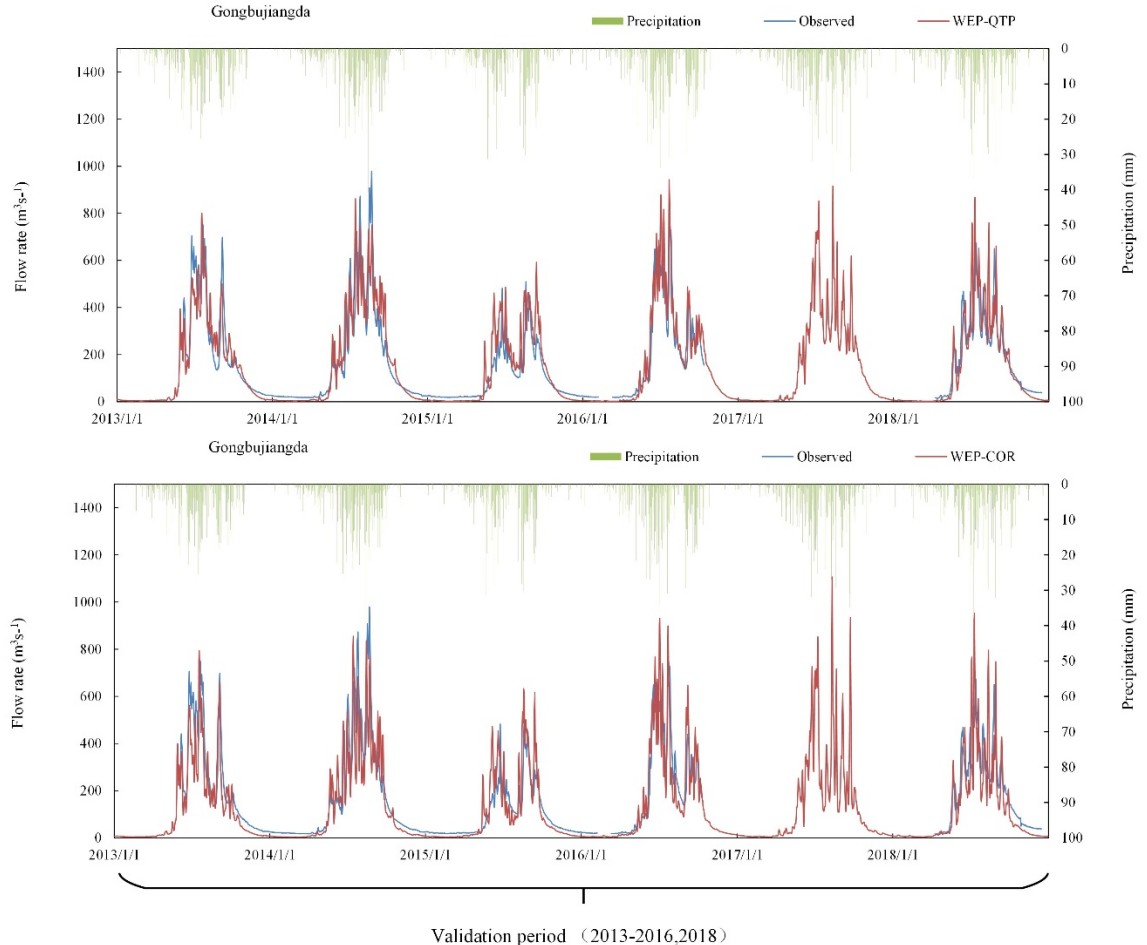

(a)



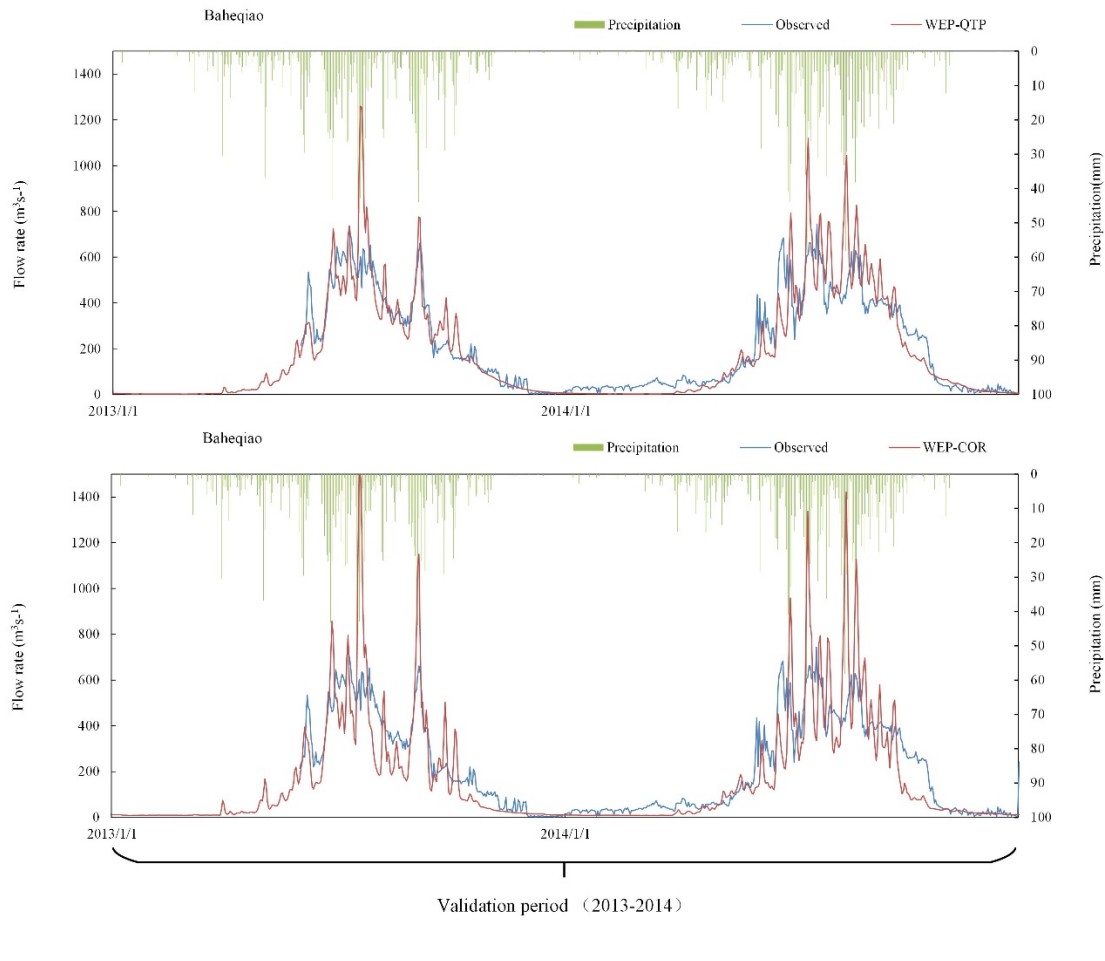

(b)





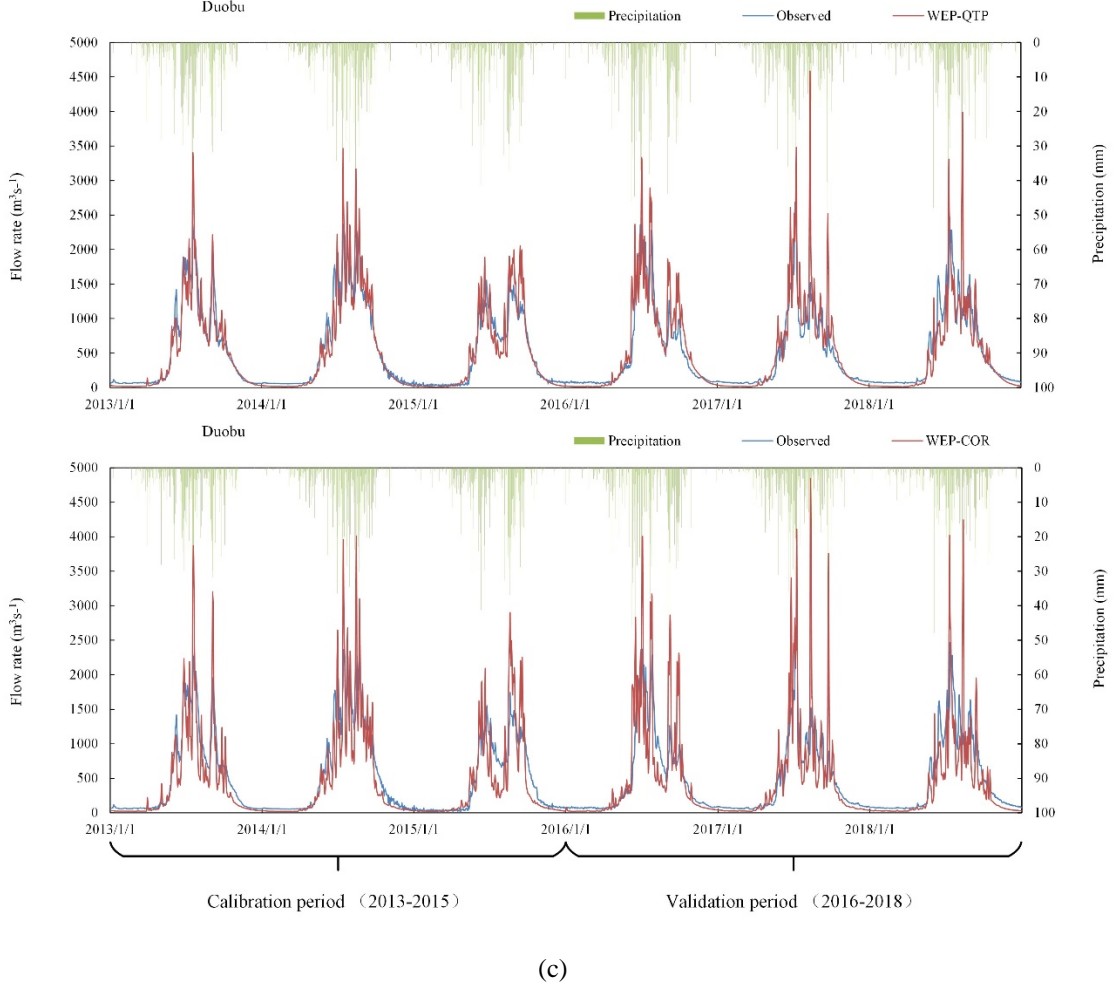

(c)

**Figure 6: Verification results of the WEP-QTP and WEP-COR models at (a) Duobu, (b) Gongbujiangda, and (c) Baheqiao stations**


### 3.2 Simulation and comparison of soil–gravel hydrothermal data at test sites

Fig.ure 7 shows the air temperature, as well as the simulated and measured snow thickness during freezing and thawing at the test point. The snow began to accumulate on 3 December 2016 and was completely melted by 4 April 2017. The maximum snow thickness was 12.4 cm, and the simulated snow thickness was consistent with the measured value. The temperature and

moisture of the soil–gravel layer at the experimental point during the freezing and thawing period of the soil were compared with the measured results.





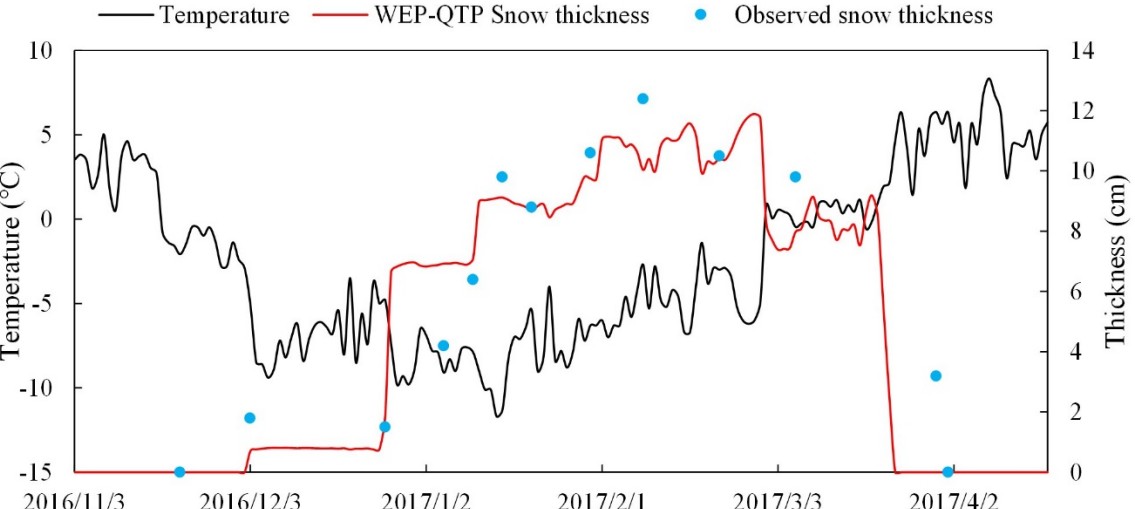

**Figure 7: Snow thicknesses and temperatures during freezing and thawing at the experimental sites from observations and the WEP-QTP model simulated results**

### 3.2.1 Soil–gravel temperature

The soil–gravel temperature simulation results of the WEP-QTP and WEP-COR models are shown in Fig. 8. The soil thickness at the test site was approximately 40 cm. For the 10 cm layer, because the simulation parameters of the WEP-QTP and WEP-COR models were the same, the temperature simulation results were consistent except during the snow cover period (i.e., the time period when the snow thickness was > 5 cm, 27 December 2016–20 March 2017). Due to the heat preservation effect of the snow, the heat transfer and temperature fluctuations of the surface soil were reduced. The RE of WEP-QTP in the 10 cm soil layer was 11.1 % during the snow cover period, which was much less than the 46.0 % of WEP-COR. The simulation results of the two models in the 20 cm layer did not differ considerably. For the 40 cm and 60 cm layers, the moisture content of the soil in the WEP-COR model was greater than that of the gravel layer in the WEP-QTP model. The higher heat capacity of water reduces the thermodynamic difference between the soil and the gravel, resulting in a small temperature difference between the WEP-QTP and WEP-COR models in the early freezing stage. However, as the temperature decreased, the moisture in the soil was converted into ice with a smaller heat capacity; thus, the difference in thermodynamic properties between the gravel and the soil gradually increased. During this period, the simulation difference between the two models reached a maximum of 1.41 °C (40 cm layer, 26 January 2017). For the temperature simulation below 60 cm, because the temperature was higher compared with that in the 40–60 cm layer, the thermodynamic properties of the gravel in the WEP-QTP model and the soil in the WEP-COR model are not significantly different due to water phase change, and the difference in the non-snow cover period was not as great as in the 40–60 cm layer. During the snow cover period, the effect of snow on the temperature





was also reduced due to the weakening of the upper soil–gravel layers. In general, the snow cover reduced the heat transfer
and temperature fluctuations of the soil layer, which improved the simulation accuracy of the surface soil temperature. In
addition, by neglecting the special hydraulic and thermodynamic properties of the gravel layer, the WEP-COR underestimated
the soil temperature. The snow and gravel layer collectively resulted in the temperature simulation difference. The average RE
of WEP-COR was −3.60 %, and that of WEP-QTP was 0.08 %. The results of the WEP-QTP simulation were closer to the
actual measurement; thus, it was able to accurately reflect the temperature changes of each layer during freeze–thaw.


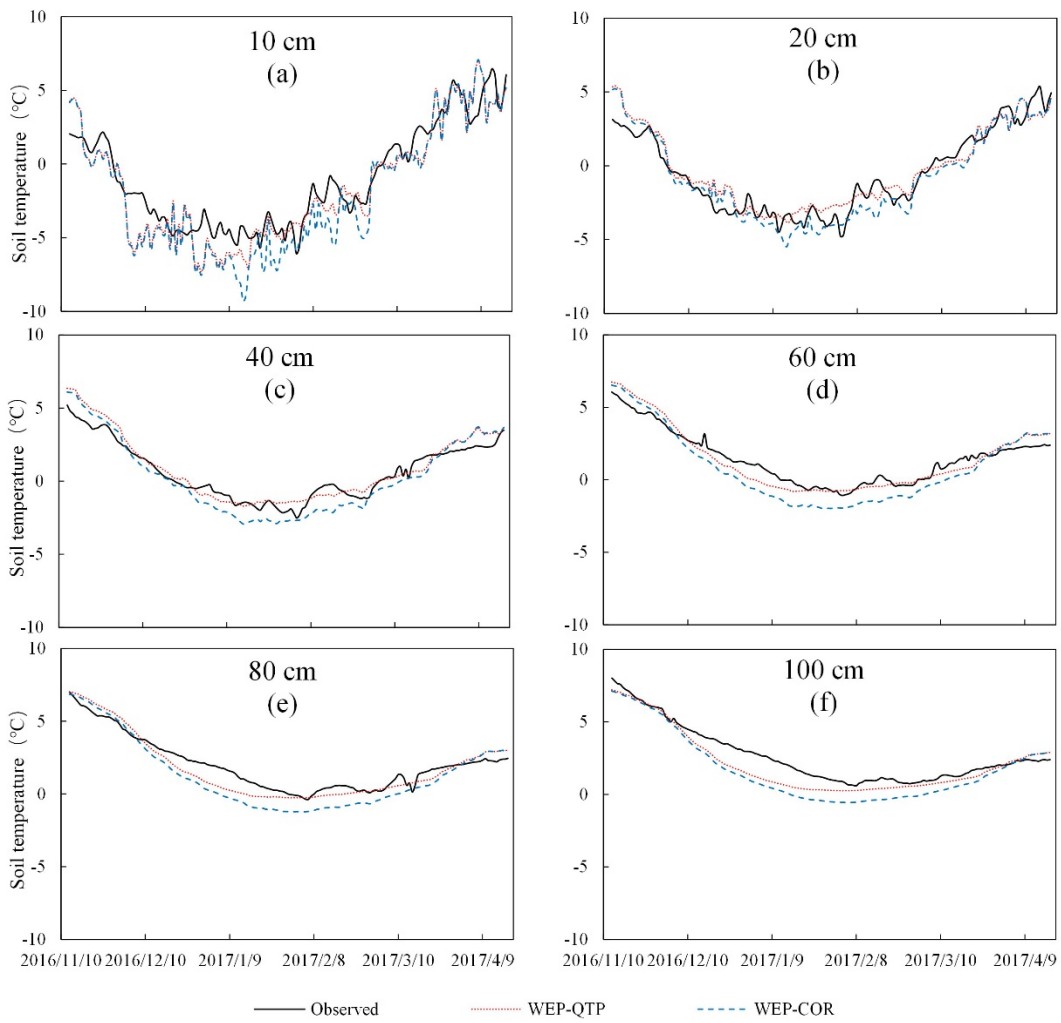

**Figure 8: Simulated (WEP-QTP and WEP-COR models) and observed temperatures of the soil–gravel layer at different depths**



### 3.2.2 Soil–gravel moisture

Figure 9 shows the comparison between the simulated and measured values of the liquid water content in the freeze–thaw period of the WEP-QTP and WEP-COR models at the experimental points in 2016–2017. During the freezing period (December–March), the upper layer liquid water content first dropped because of the temperature drop, and then the lower gravel layer liquid water content dropped, stabilizing in January–February. When the temperature increased in March, the upper layer initially began to melt, resulting in an increase in the liquid water content and then the subsequent melting of the

lower layer. After the thawing period, the upper part of the soil–gravel layer had a higher water content than that in the lower part due to the infiltration of snow melt, and it was also higher than before freezing.

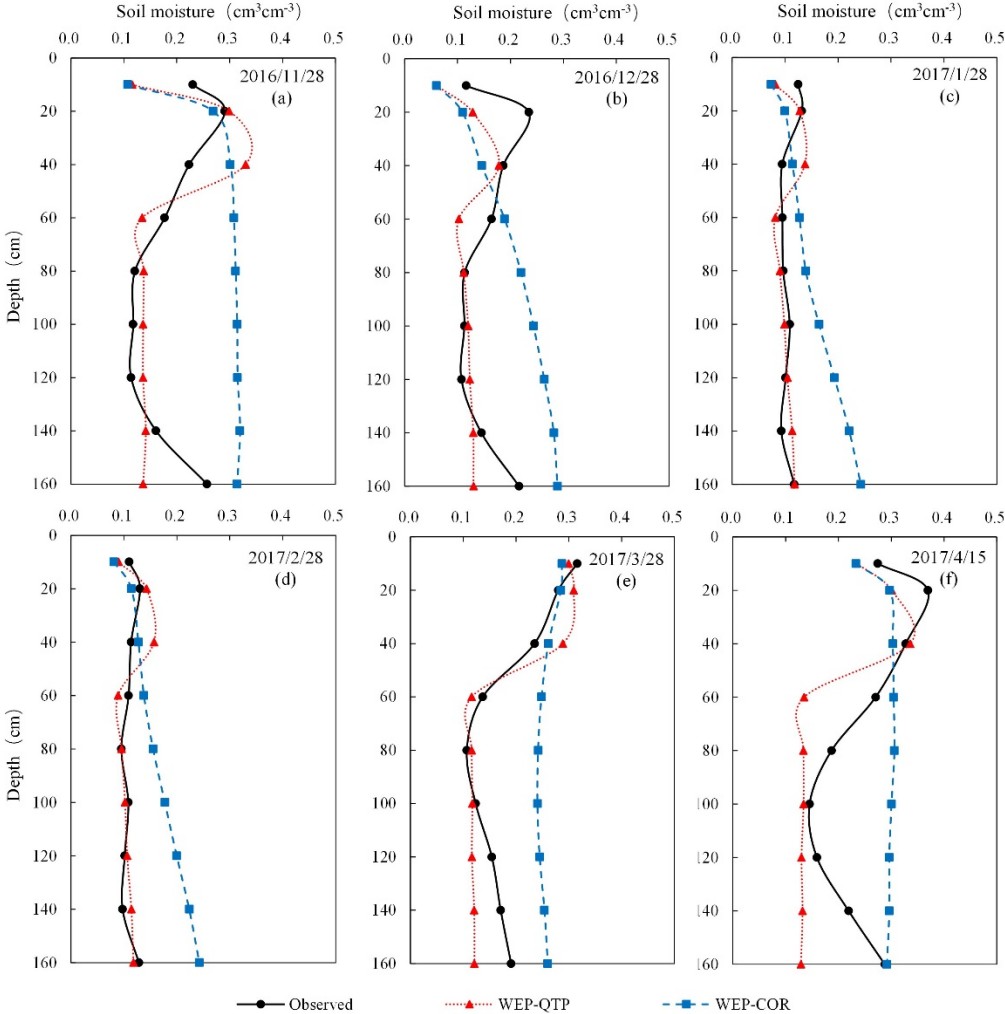



**Figure 9: Simulated (WEP-QTP and WEP-COR models) and observed moisture contents of the soil–gravel layer during the freezing**
**and thawing periods**

During the entire freeze–thaw period, the simulation of the 10 cm soil layer was less affected by the gravel layer because the water-holding capacity of the 20–40 cm soil layer was greater than that of the underlying gravel layer. The moisture in the WEP-QTP was more easily retained in the upper soil, and the moisture content was higher when the soil began to freeze and
the snow melted, which is closer to the actual measurement. Below 60 cm, the WEP-COR did not consider the impact of the gravel layer, and the water content was higher than the measured value throughout the freeze–thaw period. However, due to the large uncertainty of the compositions of the soil and gravel layer, the unstable water-holding capacity of the soil–gravel layer cannot be accurately reflected when the model is generalized, which also leads to a certain difference between the WEP-QTP simulation and the measured values. There may have been a soil interlayer at 160 cm, and the measured water content
was between the simulated values of the WEP-QTP and WEP-COR. The average RE of WEP-COR was 33.74 %, and that of WEP-QTP was smaller at −12.11 %. WEP-QTP was able to reflect the influence of gravel on the vertical migration of water.

## 3.3 Simulation and comparison of watershed flow process

To further analyze the improved WEP-QTP compared with the original WEP-COR, three sites with data for 2014 were selected, and the daily flow data of the three stations were compared with the simulated data of the two models (Fig. 10). It can be seen
from Fig. 10 that the simulation difference between the two models was mainly from June to November. The simulation performance of the WEP-QTP model was better than that of the WEP-COR model in three aspects: the peak value of the flood season was not too high; the valley value of the flow process was higher than that of the WEP-COR model; and there was a tailing process after October.



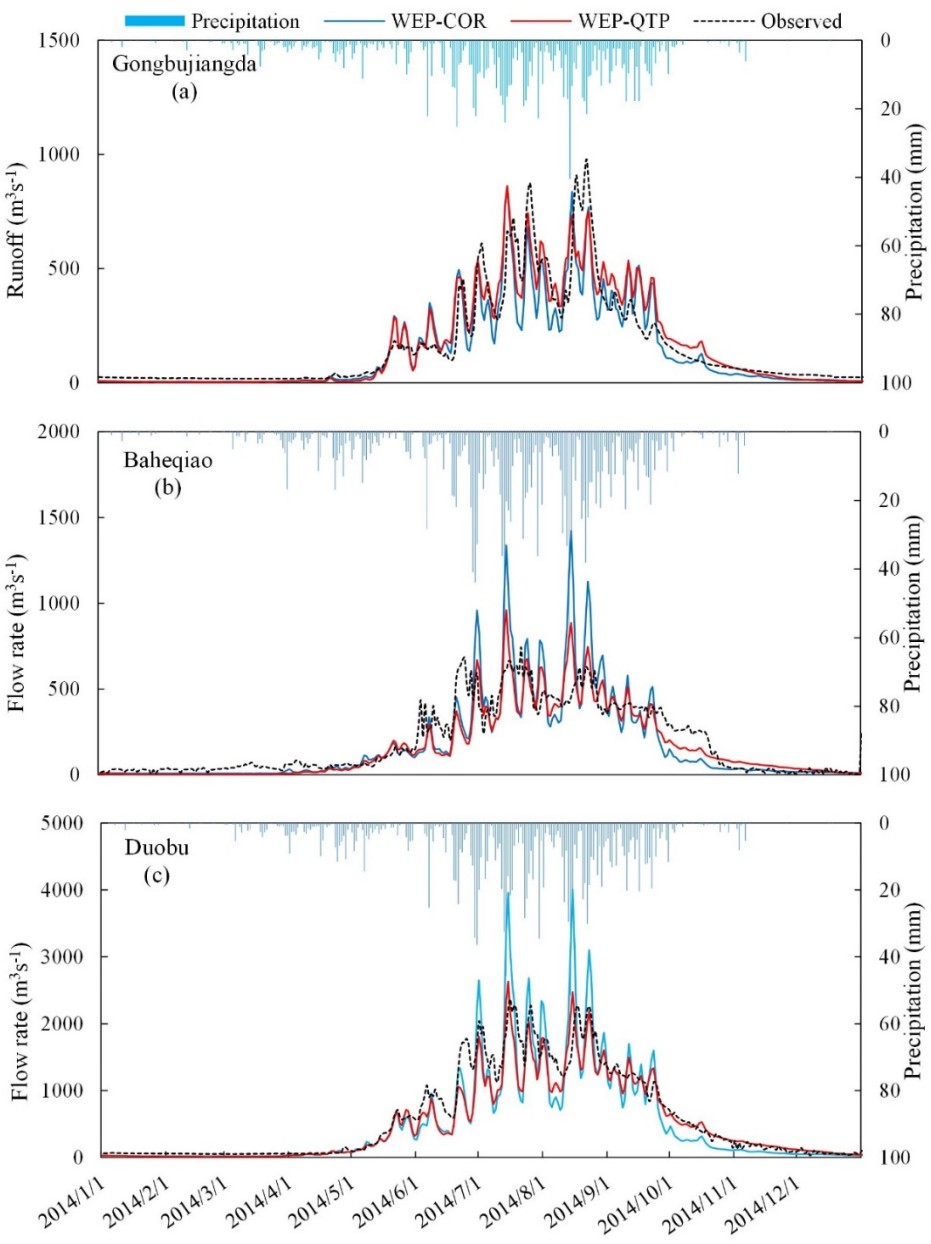

**Figure 10: Simulated (WEP-QTP and WEP-COR models) and observed flow rates at (a) Gongbujiangda, (b) Baheqiao, and (c) Duobu stations in 2014**

Figure 11 shows a comparison and analysis of the changes in hydrological cycle flux, and Fig. 12 shows the percentage of
frozen soil area in the basin. It can be seen from Fig. 11 that the runoff from rainfall of WEP-QTP was smaller than that of
WEP-COR. This was due to the fact that the WEP-QTP model can recharged groundwater more quickly during heavy rains,





and flow peaks are not easily formed. This is why the WEP-QTP model performed better in the peak simulation during the flood season. In addition, the larger saturated hydraulic conductivity of the gravel layer also increased the groundwater recharge and discharge of the WEP-QTP model, so that it performed better in simulating low values of the flow process.


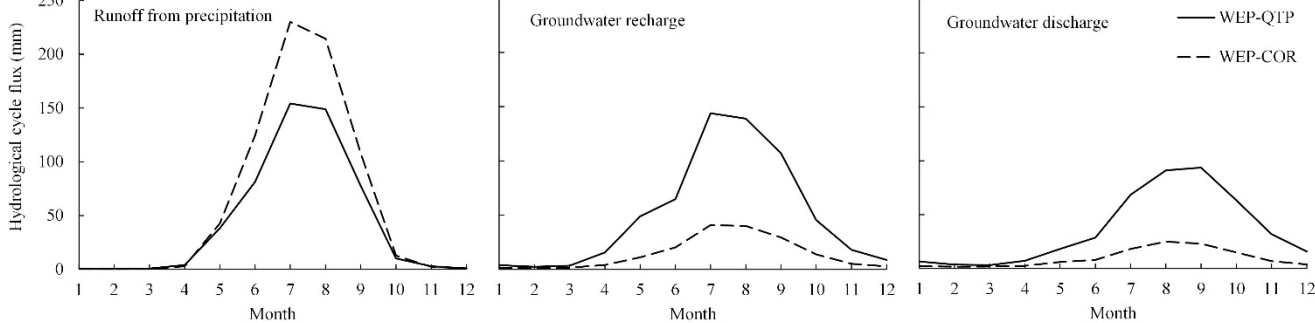

**Figure 11: Monthly change process of hydrological cycle flux in the basin**

Moreover, the temporal and spatial changes of frozen soil also affect the quantity of groundwater recharge to the river. Due to
the effect of snow and gravel, the change in frozen soil depth of WEP-QTP lagged behind that of WEP-COR; the area of WEP-COR frozen soil with a depth greater than 1 m reached its maximum in January and its minimum in August. For WEP-QTP, the value reached its maximum in March and its minimum in September (Fig. 12). This resulted in the groundwater discharge peak of WEP-QTP being one month later than that of WEP-COR (Fig. 12), and there was more time for the river groundwater recharge, which shows a better tailing process after October.


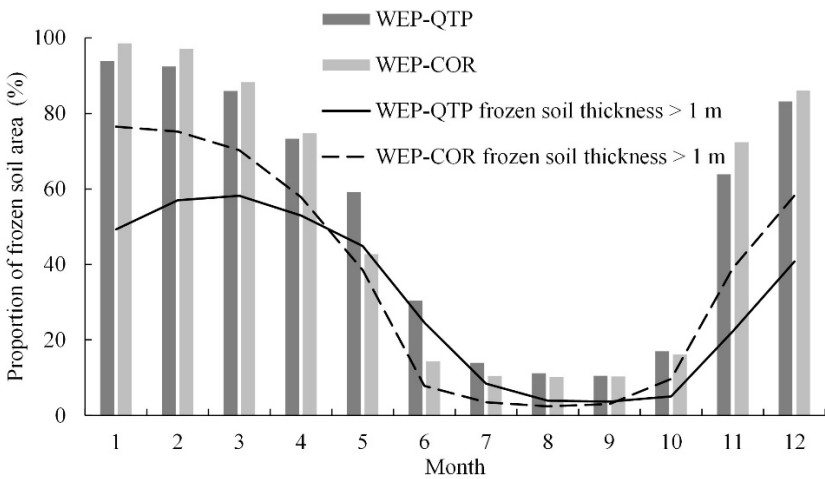

**Figure 12: Monthly change process of the frozen soil area proportion in the basin**





In general, the snow–soil–gravel layer structure changed the water circulation flux process in the basin. The peak value
simulated by the WEP-COR model was large and the valley value was small, and there may no significant difference in the
monthly runoff process. However, in the daily flow process, taking Duobu station in 2014 as an example, compared with the
measured value, the maximum difference between the RE of WEP-COR and WEP-QTP was 21.3 % (−45.0 % and −23.7 %,
respectively, on 9 August 2014) at the valley value and 88.2 % (130.5 % and 42.3 %, respectively, on 15 August 2014) at the
peak value. Ignoring the snow–soil–gravel layer structure greatly impacts the hydrological forecast, reservoir regulation, and
water resource utilization.

## 4 Conclusions

This study combined the geological characteristics of the thin soil layer on the thick gravel layer and the climate characteristics
of the long snow cover period in the Qinghai–Tibet Plateau. With the Niyang River Basin as the research area, the WEP-QTP
model was constructed based on the original WEP-COR model. This model divides the single soil structure into two types of
media: soil and gravel layers. In the non-freeze–thaw period, two infiltration models based on the dualistic soil–gravel structure
were developed based on the Richards equation in non-heavy rain periods and the multi-layer Green–Ampt model in heavy
rain periods. During the freeze–thaw period, a hydrothermal coupling model based on the continuum of the snow–soil–gravel
layer was constructed. This model was used to simulate the water cycle processes of the Niyang River Basin, and the
improvement effect of the model was analyzed by comparison with the WEP-COR model.
Compared with the simulation results prior to improvement, it was found that the addition of snow not only reduces the surface
soil temperature fluctuations, but also interacts with the gravel layer to reduce the soil freezing and thawing speed. The low
estimation of temperature by WEP-COR was corrected, and the RE was reduced from –3.60 % to 0.08 %. At the same time,
the WEP-QTP model can reflect the impact of the gravel layer under the soil on the vertical movement of water and accurately
describe the dynamic changes in moisture in the soil and gravel layers; the RE of the moisture content was reduced from 33.74 %
to −12.11 %.

According to the comparison of the WEP-QTP simulation and measured results of the main stations in the Niyang River Basin,
the daily flow process simulated by the model is in line with the actual situation, and the flow simulation result is more accurate
(Nash > 0.75 and |RE| < 10 %), which is a considerable improvement compared with the WEP-COR model. In the non-freeze–
thaw period, the dualistic soil–gravel structure increased the recharge and discharge of groundwater and improved the
regulation effect of groundwater on flow, stabilizing the water flow process. The maximum RE at the flow peak and valley
decreased by 88.2 % and 21.3 %, respectively. In the freeze–thaw period, by considering the effect of the snow–soil–gravel
layer continuum on soil freezing and thawing processes, the change in frozen soil depth of WEP-QTP lagged behind that of
WEP-COR by approximately one month. There was more time for the river groundwater recharge, which shows a better tailing
process after October.



In contrast to the general cold area, the special geological structure and climatic characteristics of the Qinghai–Tibet Plateau change the water cycle processes in the basin. Ignoring the influence of the dualistic soil–gravel structure greatly impacts the hydrological forecast and water resource assessment.

**Data and code availability**

The datasets and model code relevant to the current study are available from the corresponding author on reasonable request.

**Author contribution**

PW performed the model programming and simulations. ZZ, KW and YL conducted the field experiments. ZZ, JL, YL, and JL contributed to the model programming. The writing was performed by PW and ZZ. CX, YJ and HW contributed to the writing of the paper.

**Competing interests**

The authors declare that they have no conflict of interest.

**Acknowledgements**

This work was partly supported by grants from the National Natural Science Foundation of China (91647109) and the National Key Research and Development Program of China (2016YFC0402405).

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
