# Peer review of "Description and application of a distributed hydrological model based on soil-gravel structure in the Qinghai-Tibet Plateau"

_Hydrology and Earth System Sciences, 2021_

## Referee Comment (RC2)

General comments

This ms tried to improve the WEP-COR model by adding two parameters (gravel and snow cover), and model the water cycle on the QTP. This improved model seems reasonable and the observed results fit well with the modelled data. However, some important discussions were not fully addressed. Especially, the scientific gaps you proposed in the abstract were not presented in the discussion. In addition, the scientific hypothesis seems inappropriate. Thus, I think it is not suitable for publication in the current version. I suggest rewritten it carefully before it can be published.

Abstract

I don't think that the dominant lithology of the whole QTP is thick gravel layers. The Quaternary deposit is prevalent on the QTP. Thus, you should define the thickness of the gravel layer and the depth. If the gravel layer occurred at depth as deep as 50m or more, how can it affect the hydrological processes? Importantly, the hydrological and water cycles on the QTP differed due to the remarkable spatial heterogeneity of precipitation, the topography, and the atmospheric circulation, even in a small watershed. For example, maybe the shallow gravel layers generally occurred in the river valley and at the foot of valley, however, it may be buried in deep layer on the mountain slope. Accordingly, how can you determine the mechanism of water cycle on the entire QTP by only modelling one site? In addition, due to the occurrence of permafrost, although the high permeability of the gravel, the sub-permafrost water is hard to involve the surface

water cycle.

Line 18: If I can understand, you tried to study the water cycle mechanism of the Qinghai–Tibet Plateau via a local study in the southeast QTP, I don't think it is a good idea. As you mentioned, the geological and climatic characteristics varied on the QTP, including the distribution of cryosphere, the precipitation regimes, as well as the lithology, so how did you achieve your goal via the investigation at only one site?

non-freeze–thaw period: please define it, did you mean the absolutely freeze period? Or completely thaw period?

freeze–thaw period: the same question. How did you define it?

Introduction

1. I suggest that the authors should reconsider the hypothesis: the lithology on the QTP differed significantly, not what you said, the gravel layer only occurred in some special conditions, e.g. the low river valley or some fluvial alluvial landform. The Quaternary deposits is important. Especially when you investigate the water cycle.

2. From the title, I suggest that the first section should be focused on the importance of water cycle on the QTP. And the influence of lithology is discussed to propose the knowledge gap.

Thus, the introduction section should be rewritten.

Study area

Lack of the geological data and the lithology characteristics in this basin.

Results and discussion: the contents in this section did't fit with what you have presented in the abstract and the title.

1. As you mentioned, the ice in the embedded in the soil pore is important, so I think the ice conditions should be considered.

2. I do not see the discussions of the influence of gravel content on the model, as well as on the water cycle.

3. The discussion section is insufficient. The authors only presented the applications of model in the flow process, the moisture, etc. However, it was absolutely lack of some important things. I suggest discuss the influences of gravel layer and snow cover on the water cycle, which you aimed to address in the abstract. I think what you presented in the result section was just the model result. How you determined the water cycle using your improved model on the QTP is important.

Special suggestions

Page 2 line 42: change "permanent" to "permafrost"

Page 3 line 66-67: the geological structure of the Qinghai–Tibet Plateau is special, with a thick gravel layer under the thin soil layer. How did you draw this conclusion? As I suggested, I don't think the gravel layer is thick over the whole QTP.

Page 7 line 179: how did you set the thickness of the soil layer at different depth?

Page 8 How deep did you model?

Page 19 415-416, how did you obtained the snow cover period in this area?

Page 21 the differences between the modelled moisture by using two different models are big, how did you explain it?

Please check the grammars and expressions by inviting a native English speaker.

Page 23 line 476: I can not understand the expression" ⋯the WEP-QTP model can recharged groundwater more quickly during heavy rains⋯.". May be "⋯the groundwater recharged more quickly in the WEP-QTP model⋯"

Page 24 line 479; the same question, the expression is hard to follow, please improve the ms by inviting a native English speaker.

Page 24 line 485: what do you mean the area of WEP-COR? I think the area of frozen soil in the WEP-COR model is correct.

---

## Referee Comment (RC3)

The manuscript deals with hydrological modeling using a modified WEP-QTP distributed hydrological model, in its application to Niyang River basin in the named region. In the Abstract, early in the MS text, the authors state that their main enhancements of the original model are, (a) separating gravel layer from the 'soil' layer, whatever the soil layer is, and (b) the addition of the overland snow cover on top of the soil layer during 'freeze-thaw period'.

My overall impression is that this manuscript be rejected with encouragement to resubmit after substantial reworking.

Introduction is chaotically written and poorly referenced. Hydrological processes in permafrost environment are only vaguely explained, so that the readership might not adequately reflect on the correctness and scientific soundness of the proposed model formulation. In the future submissions, I would suggest better referencing sections concerning permafrost hydrology, show hydrology, and cold region hydrology modeling. It is unclear, quite early in the manuscript (MS), what are the 'freeze-thaw' and 'non-freeze-thaw' periods? This is unclear, because in seasonally frozen soils thaw period can be extended long into summer period, and in permafrost, phase state changes in the soil profile occur continuously.

The description of the Study region should be separated from the Materials and Methods section. In the Data description, some datasets seem to be irrelevant to the distributed model setting proposed in the MS. Rainfall stations, as noted in the description, were all situated in the river valley. I would expect, here or later in the MS, that a typical rainfall distribution over the area would be given. Also, in principle, since the WEP-QTP is a distributed model, we would need to see the distribution of major hydrologically-relevant features across the watershed, and a sort of 'hydrological response units' distribution, or subcatchments having a meaning similar to HRU conceptually.

We would need to understand how the daily precipitation signal processes across the catchment, especially where only one downstream weather station is operational in the catchment. Since the introduction of snow layer was done in the distributed model setting, we would like to know how the snow cover distribution was estimated, and how snow meltdown was assessed (in spatial terms). Typical snow cover thickness in different parts of the catchment must also be presented. Permafrost/seasonal frost distribution in the catchment is essential since it was already presented in the Introduction that permafrost affects the hydrological processes at the QTP, and it needs to be presented on a separate figure along with glacier distribution.

Experimental data were used in this research (see cf. Lines 160-166). Experimental monitoring site, program, methods and data description need to be explained, if these data are used in the MS.

In the Introduction to WEP-COR section, it is unclear whether this version of the model has already had a permafrost hydrology routine implemented or not. From Lines 190-193, I can assume this, but it is unclear how phase state transitions control water distribution across the soil layers: when soil surface is all frozen; when the residual frozen layer separates surficial soils from groundwater (seasonal frost situation).

In the Model Improvement section, 'the widespread dualistic soil–gravel structure' is once again being referred, though neither geological sections were presented nor spatial distribution of gravel thickness was given. Overall soil thickness down to transitional layer was 200 cm, first two layers were 10 cm thick and lower lying layers, 20 cm thick. What was then the transitional layer thickness? Is it constant or variable? It can be imagined from the Figure 2 that gravel layer thickness above the groundwater table could easily exceed 2 m, and reach or even exceed 5 m. No information is given in the MS concerning this particularity of the vertical model structure. Still, no explanations on how permafrost is affecting lateral routing in the subsurface. If at one point, e.g., ground temperatures were observed, these data could be presented in the respective MS section to explain permafrost/seasonal frost dynamics. It is known that clastic sediments are mostly cryotic, i.e., with low temperature and low ice content. Also, since the presence of permafrost/seasonal frost is implied, I would like to know how the convective heat exchange in gravels affects the ground temperature in the model.

In snow cover model description, was snow sublimation assessed? What was the phase separation temperature, or air temperature at which precipitation falls as snow or rain? This parameter is important for the catchment in question, since owing to the great difference in altitude, precipitation may fall as rain at lower altitudes but as snow at higher altitudes. If the model does not account for this effect, it must be stated explicitly.

For the model, it is useful to show explicitly the number of calibrated parameters, and their values: either calibrated from the model runs/sensitivity analysis or taken from the literature.

In the experimental data analysis, I am particularly surprised to see no signs of the 'zero curtain' effect during phase transitions in soils. Does this mean that soil moisture content finally is low enough to not affect the ground temperature dynamics? Also, from the experimental site data I would assume there is no permafrost, but only seasonal ground freezing, which has only limited effect on cold regions hydrology. In the same lines, I has started wondering how exactly the model treats phase state changes and latent heat release/absorbtion.

There are numerous other remarks along the text flow which I preferred to not pick up all, but rather generalize as a poor English usage, and lack of explanations/references. I do not believe the manuscript can be published in HESS in the present form.

---

## Author Comment (AC3)

*The manuscript modified the a distributed hydrological model based on soil–gravel structure, and applied it to a watershed in the QTP. The topic fell into the scope of this journal and there are some issues need to be addressed as shown below:*

Dear reviewer:

We appreciate the detailed and valuable comments, which have considerably improved the quality of our manuscript. Our responses to the comments are provided below.

*About the novelty of the study. Considering the soil layer and unconfined aquifer in the cold region and QTP have been test for some previous studies, the author need to compare the current study with some previous studies. For example:*

*Song L., Wang L., Li X., et al. Improving permafrost physics in a distributed cryosphere-hydrology model and its evaluations at the upper yellow river basin. Journal of Geophysical Research: Atmospheres, 2020, 125(18):1-22.*

*Sun, A., Yu, Z., Zhou, J., Kumud,A., Ju, Q., Xing, R, Huang, D, Wen, L., 2020. Quantified hydrological responses to permafrost degradation in the headwaters of the Yellow River (HWYR) in High Asia. Sci. Total Environ. 712, 135632.*

*Gao B., Yang D., Qin Y., et al. Change in frozen soils and its effect on regional hydrology, upper Heihe basin, northeastern Qinghai-Tibetan Plateau, Cryosphere, 2018, 12(8):657-673*

Reply: Thanks for the comment and suggestion. We have carefully studied the previously published work on the influence of frozen soil on the hydrological cycle in the QTP including those introduced by reviewer. The survey revealed that most of those studies define the simulated object of hydrothermal coupled transport process as a one-dimensional homogeneous medium. However, the geological features of the QTP are generally thin soil layers above the thick gravel layers, and there are clear boundaries between them. Compared with soil, gravel layers have different porosity and density as well as water and thermal properties, which have a greater effect on hydraulic conductivity and water retention curve. Adjusting these parameters with the proportion of gravel can improve the hydrothermal simulation effect to a certain

extent. However, owing to the generalization of soil into a one-dimensional homogeneous structure, it is difficult to reflect the influence of this region's obvious upper and lower layered geological structure on hydrothermal migration and watershed hydrological cycle only through parameter adjustment. Based on the general geological characteristics of the QTP, in this study, we generalized the hydrothermal coupling simulation object into the binary medium of soil and gravel. Combined with the characteristics of hydrothermal transport in different periods, two methods were used to simulate hydrothermal transport in freeze-thaw period and non-freeze-thaw period, which improved the accuracy of hydrothermal simulation. These details of theoretical and methodological differences from previous studies will be supplemented in the Introduction and Discussion in the revised version.

*For the study area, the author need to describe the distribution of frozen soil. Where is the permafrost and seasonally frozen ground? For the experiment site, is it in permafrost region or seasonally frozen ground?*

Reply: Thanks for the comment. The frozen soil in the study area is mainly seasonal. Permafrost accounts for approximately 23.65%, mainly distributed in the upper reaches of the basin and the high-altitude areas on both sides of the main stream. The annual average temperature of the experimental site is 5.28℃, which is a seasonally frozen soil area. In the revised manuscript, we will supplement these details in the section 2.1.1 Study area.

*For figure 3, how do you determine the thickness of each soil layers? What is the maximum frozen depth of the study area? Do you consider the freezing front when you divide the soil layer?在永久冻土区，我们假设不透水层为*

Reply: Thanks for professional questions, we will make it clearer here and in the revised manuscript. The division of soil layer thickness was mainly based on the convenience of accurately simulating the hydrothermal migration process. As the surface soil is more sensitive to changes in atmospheric temperature, the thicknesses

of the first and second layers were set to 10 cm each, and the third through eleventh layers were divided evenly, each with a thickness of 20 cm. The depth from the surface to the impermeable layer was an adjustable parameter in the model that can be adjusted based on the actual basin conditions. Considering the active range of seasonal frozen soil, the depth of the hydrothermal numerical simulation in the model was set 2 m. When the groundwater depth was greater than 2 m, the excess comprised the transition layer. The maximum frozen soil depth was located in the permafrost regions. In these regions, the lower permafrost layer was used as the lower boundary condition of the model. The maximum frozen depth was the depth from the surface to the impermeable layer. The position of the freezing front was determined using the interpolation of the layer thickness and temperature at the center of the upper and lower layers. We have redrawn Figure 4 to ensure that the readers can better understand our model structure.

[Figure]

Figure 4. Layered structure of the dualistic "soil–gravel" structure

*Eq 14, I suggest to give the equations about how to calculate LE and H.*

Reply: We will add that the *LE* is related to the sublimation and evaporation rates and can be obtained as follows:

$$LE = L_{vi}E_{subl} + L_{vl}E_{evap} \approx L_{vi}E$$

where, $L_{vi}$ is the latent heat of sublimation of ice ($2.838 \times 10^6$ J/kg at 0°C); $L_{vl}$ is

the latent heat of evaporation of water (2.505 × 10⁶ J/kg at 0 °C); $E$ is the sum of the surface sublimation and evaporation rates.

$H$:

$$H = \frac{\rho_a C_P (T_s - T_a)}{r_a}$$

where, $\rho_a$ is the air density; $C_P$ is the constant pressure specific heat of the air; $T_s$ is the surface temperature; $T_a$ is the air temperature; and $r_a$ is the aerodynamic impedance.

The above formula was combined with the ground heat conduction equation and energy balance equation and solved using the iterative method, which requires extensive calculations. Therefore, in this study, we approximately deduced the heat into the ground according to the daily temperature change, and simplified the calculation by solving the $H$ according to the energy balance equation after calculating the $LE$.

***Are Supra-permafrost water and Sub-permafrost water both exit in the study area?***

Reply: The supra-permafrost water and sub-permafrost water are important components of the hydrological cycle in the cold region. We will make it clearer in the revision that the sub-permafrost water is generally confined, and the interaction between the sub-permafrost water and the hydrological process is weak owing to the impermeability of the permafrost layer. In this study, for the permafrost, the freezing and thawing processes of the surface layer were considered, and the lower permafrost layer was used as the lower boundary condition. We simplified this process and only considered the exit of supra-permafrost water.

***The radiation transfer in the snow layer are ignored in this study, the author may discuss the uncertainty from this? Another question, how do you estimate the snow albedo to get the net radiation of snow surface?***

Reply: Thanks for the insightful comment. Radiation will get into the snow and

participate in the water phase transition and heat transfer process. However, owing to the high reflectivity of the snow to short-wave radiation, the short-wave radiation into the snow is much less than long-wave radiation into the snow. Considering that the main factors affecting the long-wave radiation are the temperature of the atmosphere and snow cover, in this study, we used the temperature index method to simplify the simulation of the snowmelting process. The value of the temperature index was determined by parameter calibration to characterize the effect of temperature and radiation on the snowmelting process. In this model, the snow albedo was fixed at 0.8. We will clarify it better in the revised version.

*How do you calibrate the parameters of the hydrological model? And what are the major parameters you calibrated?*

Reply: Sorry for the unclearness on this part. We will make it clearer here as well as in the revised version that the parameters of the model were divided into four categories: underlying surface parameters, vegetation parameters, soil parameters, and aquifer parameters. All parameters have physical meaning and can be estimated based on observational experimental data or remote sensing data. The sensitivity of the above four types of parameters was analyzed, and the sensitivity of these parameters was divided into three levels: high, medium, and low. Only the high-sensitive parameters, i.e. soil thickness, soil saturated hydraulic conductivity, and riverbed material permeability coefficient were calibrated using the daily flow process data of Duobu stations from 2013 to 2015 with Nash Sutcliffe efficiency (NSE) and relative error (RE) as the objective functions..

*There seems an underestimation of river discharge by the WEP-QTP in the freeze season, why?*

Reply: We will make it clearer that the underestimation of river discharge in the freeze season can be attributed to the following two reasons: the geological conditions in the cold plateau region are complex, and the runoff process was not fully reflected by the model because the model simplifies the groundwater simulation. In addition, the main

focus of this study was the aquifer above the impervious boundary. The outflow of sub-permafrost water was not explicitly considered, and this part of water can also supplement the river discharge in the freeze season through the macropores in the bedrock fracture zone.

***It seems that at 20 cm, the variation of soil temperature of WEP-QTP is reduced and the variation is lower than observations and WEP-COR, why?***

Reply: Considering the heat preservation effect of the snow, the soil temperature variation of WEP-QTP in the 20 cm soil layer was less than that of WEP-COR.

The thermodynamic parameters of snow cover in this study were a function of snow density. In the model, the snow density increases with the decrease of temperature, unlike the snow density in the physical model, which only increases with snow thickness and degree of melting. The thermal conductivity of snow decreases with decreasing temperature during soil freezing, resulting in the smaller variation of soil temperature of WEP-QTP, which needs to be further improved.

***I suggest to show the comparison of long term changes in the simulated runoff in the winter and summer and spring by different models and observations.***

Reply: We thank you for your valuable suggestion; we are also working on a long-term runoff evolution analysis. However, owing to the limited space, in this study we mainly considered the influence of snow and gravel on hydrothermal transport in the freeze-thaw period and the non-freeze-thaw period to construct a hydrothermal multi-structure simulation method for the underlying surface of the plateau cold region. The effects of the special underlying surface conditions of the plateau on the main water cycle processes such as direct surface runoff, groundwater recharge, and groundwater recharge channels were analyzed. In the ongoing work, we will analyze the evolution law of long-term variation of seasonal runoff and its components in the basin from various elements such as meteorology and vegetation.

---

## Author Comment (AC4)

*General comments*

*This ms tried to improve the WEP-COR model by adding two parameters (gravel and snow cover), and model the water cycle on the QTP. This improved model seems reasonable and the observed results fit well with the modelled data. However, some important discussions were not fully addressed. Especially, the scientific gaps you proposed in the abstract were not presented in the discussion. In addition, the scientific hypothesis seems inappropriate. Thus, I think it is not suitable for publication in the current version. I suggest rewritten it carefully before it can be published.*

Dear reviewer:

We appreciate the detailed and valuable comments, which will substantially improve the quality of our manuscript. We will rewrite the manuscript carefully following the comments and suggestions provided by the three reviewers in the revised version. Our responses to the comments are provided below.

*Abstract*

*I don't think that the dominant lithology of the whole QTP is thick gravel layers. The Quaternary deposit is prevalent on the QTP. Thus, you should define the thickness of the gravel layer and the depth. If the gravel layer occurred at depth as deep as 50m or more, how can it affect the hydrological processes? Importantly, the hydrological and water cycles on the QTP differed due to the remarkable spatial heterogeneity of precipitation, the topography, and the atmospheric circulation, even in a small watershed. For example, maybe the shallow gravel layers generally occurred in the river valley and at the foot of valley, however, it may be buried in deep layer on the mountain slope.*

*Accordingly, how can you determine the mechanism of water cycle on the entire QTP by only modelling one site? In addition, due to the occurrence of permafrost, although the high permeability of the gravel, the sub-permafrost water is hard to involve the surface water cycle.*

*Line 18: If I can understand, you tried to study the water cycle mechanism of the Qinghai–Tibet Plateau via a local study in the southeast QTP, I don't think it is a good idea. As you mentioned, the geological and climatic characteristics varied on the QTP, including the distribution of cryosphere, the precipitation regimes, as well as the lithology, so how did you achieve your goal via the investigation at only one site?*

*non-freeze–thaw period: please define it, did you mean the absolutely freeze period?*

***Or completely thaw period? freeze–thaw period: the same question. How did you define it?***

Reply: Thanks for the insightful comments. Sorry for being not clear enough in the previous version of the ms on the aspects pointed out by the reviewer, which will be clarified here as well as in the revised version of the manuscript. As the reviewer said, the Quaternary deposit is prevalent on the QTP. The focus of this study was the hydrothermal migration process in the Quaternary deposits (including the deposits in the upper layer that have developed into soil and the mixture of sand and gravel, and soils in the lower layer that have not yet been fully developed), rock layer below the Quaternary sediments was not within the scope of this study.

Affected by topography and atmospheric circulation, although there are regional differences in the water cycle process of the QTP, there also share certain similarities. During the continuous uplift of the QTP, a series of ascending areas (denuded areas) and descending areas (deposited areas) have been formed in this area. Quaternary deposits are generally thinner in denuded areas and thicker in deposited areas (valleys, plain). The thickness of Quaternary deposits varies greatly at the transition between the denuded areas and the deposited areas controlled by the fault zone. The soil forming process of Quaternary deposits on the QTP is intermittently affected by surface uplift. In addition, under strong freeze-thaw conditions in the cold plateau region, the humus accumulation of herbaceous plants is slow and the decomposition of minerals is weak, resulting in slow soil development and thin soil layers. This phenomenon is prevalent in the QTP (Sun, 1996; Yang et al., 2009; Chen et al., 2015). Moreover, we also confirmed this phenomenon in the present study through field campaigns in the study area (field investigation, sampling, laboratory analysis, etc.). In view of this phenomenon, most of the previous studies have improved the accuracy of hydrothermal simulation by adjusting the parameterization scheme. However, because the soil was generalized into a homogeneous structure, it was difficult to reflect the influence of this region's obvious upper and lower layered geological structure on hydrothermal migration and on watershed hydrological cycle only through parameter adjustment.

Based on this geological feature of the QTP, in this study, we took the aquifer and unsaturated zone above the impervious boundary as the research object (this part contains both soil layer and gravel layer, with a strong freeze-thaw effect and close interaction with surface hydrological process, which is the key link affecting the hydrological cycle in this area). Through field experiments in the Niyang River Basin, the simulation object of hydrothermal coupling was generalized as the binary medium of soil and gravel. Considering the characteristics of hydrothermal migration in

different periods, the simulation of hydrothermal transport was carried out by dividing the freeze-thaw period and the non-freeze-thaw period. On this basis, the influence of the ubiquitous geological structure of the QTP on the hydrological cycle of the basin was analyzed.

The lithosphere is basically below the Quaternary deposits or deeper below the permafrost. Affected by the impermeability of rock and permafrost, the interaction between this part and the hydrological process is weak, which was not considered in this study.

For other differences such as glacier distribution, precipitation and climate characteristics, in this study, we fully considered the effect of these differences on hydrological simulations through glacier remote sensing and spatial interpolation of precipitation and temperature in the Niyang River Basin. The spatial differences of these hydrological elements on a larger scale in the QTP need to be further studied by combining remote sensing and spatial downscaling.

The non-freeze-thaw period defined in this model was the complete thawing period. We will also make it clearer that in this study, the division between the freeze–thaw and non-freeze–thaw periods was determined according to the soil and gravel temperatures in each calculation unit. Non-freeze–thaw period is defined as when all the soil/gravel layer temperatures were greater than 0 °C, and all of the water was a liquid. As long as a layer of soil or gravel had temperature below 0 °C, it is considered as freeze–thaw period.

**Introduction**

**1. *I suggest that the authors should reconsider the hypothesis: the lithology on the QTP differed significantly, not what you said, the gravel layer only occurred in some special conditions, e.g. the low river valley or some fluvial alluvial landform. The Quaternary deposits is important. Especially when you investigate the water cycle.***

Reply: Thanks for the comments. For more details please refer to our reply to the comments on Abstract above. The soil and gravel structures are common in this area. The gravel layer does not occur only in some special conditions. The structure above the impervious boundary has been summarized in the figure below:

[Figure]

The object of this study was the hydrothermal migration process in the Quaternary deposits (including the deposits in the upper layer that have developed into soil and the mixture of sand, gravel, and soils in the lower layer that have not yet been fully developed), rock layer below the Quaternary sediments was not within the scope of this study. We will redefine this concept in the Introduction.

**2. *From the title, I suggest that the first section should be focused on the importance of water cycle on the QTP. And the influence of lithology is discussed to propose the knowledge gap. Thus, the introduction section should be rewritten.***

Reply: Our writing ideas and the reviewer's suggestions are the same, sorry that we failed to express it clear enough in the original version. In the revised version we will make it clearer that Paragraph 1 introduces the importance of the QTP to water resources in China and Southeast Asia. Paragraph 2 illustrates the effect of frozen soil on the water cycle in the region. The rest paragraphs highlight the knowledge gap by comparing with the frozen soil hydrology research that has been carried out in the QTP, state the research objectives and implications. We will supplement the relevant information and revise the Introduction in the revised manuscript.

*Study area*

***Lack of the geological data and the lithology characteristics in this basin.***

Reply: Thank you for pointing out this deficiency. The Yarlung Zangbo sediments consist of greywacke and litharenitewith low-moderate maturities of textures and minerals (Huyan et al., 2022). In the Niyang River Basin, where the study was carried out, we measured the basic geological conditions in the basin by field campaigns. We will supplement this information in Section 2.1.1 in the revised version.

***Results and discussion***

*The contents in this section did't fit with what you have presented in the abstract and the title.*

Reply: Thanks for the insightful comment. In the Abstract, we will reformulate the purpose of the study (Lines 17-20) as follows:

In order to deeply study the influence mechanism of the underlying surface structure on the hydrothermal migration and water circulation process in the Qinghai–Tibet Plateau, we performed comprehensive study combining field experiments of the water and heat transfer processes and development of a Water and Energy transfer Processes model in the Qinghai–Tibet Plateau (WEP-QTP), based on the original Water and Energy transfer Processes model in Cold Regions (WEP-COR). The Niyang River Basin located on the Qinghai–Tibet Plateau was selected as the study area to evaluate the agreement between theoretical hypothesis, observation and modeling results.

In addition, we have also revised the title as "Application of a new distributed hydrological model based on soil–gravel structure in the Niyang River Basin, Qinghai–Tibet Plateau" to better reflect our research content.

**1. *As you mentioned, the ice in the embedded in the soil pore is important, so I think the ice conditions should be considered.***

Reply: Thanks for the comment. In this study, the effect of ice in soil pores on the water cycle was manifested as the effect on soil saturated hydraulic conductivity (Equation 27). When the temperature of the soil or gravel layer is lower than 0°C, the water in the pores freezes, blocking the pores and affecting the soil saturated hydraulic conductivity. Considering the difference in the hydrodynamic properties of the soil and gravel layer, different values of $K_0$ were used in the soil and gravel layer.

**2. *I do not see the discussions of the influence of gravel content on the model, as well as on the water cycle.***

Reply: The effect of gravel content on the hydraulic properties of the gravel layer was shown in Equation 3. Through field sampling in the Niyang River Basin, we found that the gravel content in the gravel layer in the basin is 50%–65% (Line140), and the gravel content in the model was determined and adjusted according to the measured value.

**3. *The discussion section is insufficient. The authors only presented the applications of model in the flow process, the moisture, etc. However, it was absolutely lack of some important things. I suggest discuss the influences of gravel***

*layer and snow cover on the water cycle, which you aimed to address in the abstract.*
*I think what you presented in the result section was just the model result. How you*
*determined the water cycle using your improved model on the QTP is important.*

Reply: Thanks for the comment. In the revised version, in addition to the comparative analysis of the simulation results in Sections 3.1 and 3.2, we will enhance the discussion on the effects of gravel layers and snow cover on the water cycle in Section 3.3. Snow and gravel alter the water cycle process by affecting the temporal and spatial changes in frozen soil. In terms of water transfer, the large pores of the lower gravel layer increase the regulation and storage effect of groundwater of the flow processes. For heat transfer, snow and gravel block heat conduction, slow down the freezing and thawing rates of aquifers, and thus change the water circulation flux process (Fig.12). In the revised manuscript, we will also further discuss the similarities and differences between our study and the previous hydrological model studies in the QTP.

*Special suggestions*

*Page 2 line 42: change "permanent" to "permafrost"*

We have made the necessary revisions throughout the ms.

*Page 3 line 66-67: the geological structure of the Qinghai–Tibet Plateau is special, with*
*a thick gravel layer under the thin soil layer. How did you draw this conclusion? As I*
*suggested, I don't think the gravel layer is thick over the whole QTP.*

Reply: Thanks for the comment. Referring to our detailed response to the comments on Abstract, we have improved the description of this important part by combining our field experiments results of the soil texture and the common consensus in literature (Sun, 1996; Yang et al., 2009; Chen et al., 2015). This phenomenon is prevalent in the QTP.

*Page 7 line 179: how did you set the thickness of the soil layer at different depth?*

Reply: Thanks for professional questions, and sorry for being *unclear*. Similar comment has also been raised by Reviewer #1. The division of soil layer thickness was mainly based on the convenience of accurately simulating the hydrothermal migration process. As the surface soil is more sensitive to changes in atmospheric temperature, the thicknesses of the first and second layers were set to 10 cm each, and the third through eleventh layers were divided evenly, each with a thickness of 20 cm.

*Page 8 How deep did you model?*

Reply: The depth from the surface to the impermeable layer was an adjustable parameter in the model that can be adjusted based on the actual basin conditions. Considering the activity range of seasonal frozen soil, the depth of the hydrothermal numerical simulation in the model was 2 m. When the groundwater depth was greater than 2 m, the excess comprised the twelfth layer (transition layer). We have redrawn Figure 4 to ensure that the readers can better understand our model structure.

[Figure]

Figure 4. Layered structure of the dualistic "soil–gravel" structure

***Page 19 415-416, how did you obtained the snow cover period in this area?***

Reply: The model constructed includes snow accumulation and melting process, and the thickness of the simulated snow at the experimental point was calibrated with the measured value (Fig.7). The snow cover period was decided according to the snow thickness simulated by the model. When the snow thickness was > 5 cm, it was the snow cover period.

***Page 21 the differences between the modelled moisture by using two different models are big, how did you explain it?***

Reply: The WEP-COR model ignores stratified geological features and the simulation object was homogeneous soil. Therefore, the simulated moisture changes gradually in the vertical direction, and there was a large difference from the measured value below the 40 cm layer (the soil layer thickness at the experimental point is 40 cm, and the gravel layer (mixed layer of gravel and soil) appears below 40 cm.). The WEP-QTP model took this geological feature into account. The existence of gravel increases the hydraulic conductivity of the gravel layer and reduces its water retention capacity, which is manifested in the simulated difference of the water content. The results of the improved WEP-QTP model were closer to the measured values. A detailed description of this

difference can be found in lines 452–459 of the current manuscript.

*Please check the grammars and expressions by inviting a native English speaker.*

*Page 23 line 476: I can not understand the expression" …the WEP-QTP model can recharged groundwater more quickly during heavy rains….". May be "…the groundwater recharged more quickly in the WEP-QTP model…"*

*Page 24 line 479; the same question, the expression is hard to follow, please improve the ms by inviting a native English speaker.*

Reply: Sorry for the grammar error, before the revised version is submitted, we will invite native English speakers to polish the language.

*Page 24 line 485: what do you mean the area of WEP-COR? I think the area of frozen soil in the WEP-COR model is correct.*

Reply: Thanks, we will correct it and other potential errors native English speakers in the revised version.

**References**

[1] Sun H L, The formation and evolution of the Qinghai-Tibet Plateau, China: Shanghai Science and Technology Press, 1996,ISBN: 9787532340231.

[2] Yang K, Chen Y Y, Qin J. Some practical notes on the land surface modeling in the Tibetan Plateau[J]. Hydrology and Earth System Sciences, 2009, 13(5): 687-701.

[3] Chen H, Nan Z, Zhao L, et al. Noah modelling of the permafrost distribution and characteristics in the West Kunlun area, Qinghai-Tibet Plateau, China[J]. Permafrost and Periglacial Processes, 2015, 26(2): 160-174.

[4] Huyan Y, Yao W, Xie X, et al. Provenance, source weathering, and tectonics of the Yarlung Zangbo River overbank sediments in Tibetan Plateau, China, using major, trace, and rare earth elements[J]. Geological Journal, 2022, 57(1): 37-51.

---

## Author Comment (AC5)

The manuscript deals with hydrological modeling using a modified WEP-QTP distributed hydrological model, in its application to Niyang River basin in the named region.

In the Abstract, early in the MS text, the authors state that their main enhancements of the original model are, (a) separating gravel layer from the 'soil' layer, whatever the soil layer is, and (b) the addition of the overland snow cover on top of the soil layer during 'freeze-thaw period'. My overall impression is that this manuscript be rejected with encouragement to resubmit after substantial reworking.

**Dear reviewer:**

We appreciate your detailed and valuable comments, which have helped us to considerably improve the quality of the manuscript. Our point-by-point responses to the comments are provided below. We hope that the revisions will sufficiently address the shortcomings of the previous version of the manuscript.

**Introduction is chaotically written and poorly referenced. Hydrological processes in permafrost environment are only vaguely explained, so that the readership might not adequately reflect on the correctness and scientific soundness of the proposed model formulation. In the future submissions, I would suggest better referencing sections concerning permafrost hydrology, show hydrology, and cold region hydrology modeling.**

Reply: Thanks for the advice. It is indeed that the Qinghai-Tibet Plateau includes both permafrost and seasonally frozen soil areas. The water and heat transfer processes related to permafrost/seasonally frozen soil are important parts of the water cycle on the Qinghai-Tibet Plateau. Seasonally frozen soil thaws completely during the summer, while only the surface layer of permafrost thaws during the summer. The lower permafrost layer remains frozen surface water and groundwater, and its freezing and thawing processes are affected considerably by the geological structure, which was the focus of this study. In regions with seasonally frozen soil, we considered the entire freezing and bidirectional thawing processes of the soil–gravel layer. For the permafrost regions, the surface layer freezing and thawing processes were considered, while the lower permafrost layer was used as the lower boundary condition. In permafrost areas, supra-permafrost water, an impermeable interlayer, or other

conditions may also be present, thereby complicating the situation. For these areas, we simplified the simulation process to mainly address the hydrothermal transport in the active layer within 2 m of the surface layer.

We agree that in the introduction, the description of the impacts of seasonally frozen soil and permafrost on the water cycle on the Qinghai-Tibet Plateau wasn't adequate enough to explain the hydrological processes in the permafrost environment. In the revised version, emphasis will be paid to referencing sections concerning permafrost hydrology, show hydrology, and cold region hydrology in the region, and to further clarify the research objectives, and to highlight the impact of scientific issues in cold plateau regions.

It is unclear, quite early in the manuscript (MS), what are the 'freeze-thaw' and 'non-freeze-thaw' periods? This is unclear, because in seasonally frozen soils thaw period can be extended long into summer period, and in permafrost, phase state changes in the soil profile occur continuously.

Reply: Thank you for this comment. We will make it clearer that in this study, the division between the freeze-thaw and non-freeze-thaw periods was determined according to the soil and gravel temperatures in each calculation unit. Non-freeze-thaw period is defined as when all the soil/gravel layer temperatures were greater than 0 °C, and all of the water was a liquid. As long as a layer of soil or gravel had temperature below 0 °C, it is considered as freeze-thaw period. Due to the undulating terrain in the study area, the temperature range in the basin is large. Different regions may be in different periods at the same time. The freeze-thaw period may extend throughout the entire year in regions with lower average annual temperatures, while regions with higher average annual temperatures may have freeze-thaw periods that only occur during the winter and spring. We will add this information to the introduction.

**The description of the Study region should be separated from the Materials and Methods section. In the Data description, some datasets seem to be irrelevant to the distributed model setting proposed in the MS.**

Reply: Thank you for pointing this out. In the revised version, we will reorganize the structure

of the manuscript accordingly, by dividing the Materials and Methods section into two sections: "Study area and data" and "Methodology".

The Data description section will also be carefully revised to correctly and clearly present all the data used in the study, including their types, quality, resolutions, period, and sources.

Rainfall stations, as noted in the description, were all situated in the river valley. I would expect, here or later in the MS, that a typical rainfall distribution over the area would be given. Also, in principle, since the WEP-QTP is a distributed model, we would need to see the distribution of major hydrologically-relevant features across the watershed, and a sort of 'hydrological response units' distribution, or subcatchments having a meaning similar to HRU conceptually.

Reply: Thank you for this comment. Contour maps of annual precipitation (Fig. a) and annual mean temperature (Fig. b) in the Niyang River Basin are given below. Figures of other data, including land type and vegetation coverage, among others, will be added to the supplementary materials in the revised manuscript.

Figure a. Annual precipitation contour map (mm/year).

Figure b. Mean annual temperature (°C).

We would need to understand how the daily precipitation signal processes across the catchment, especially where only one downstream weather station is operational in the catchment.

Reply: Two weather stations were used for the spatial interpolation of precipitation: Nyingchi station (in the downstream area) and Jiali station (outside the upstream basin) (Fig. 1 in the MS). In addition, we also obtained precipitation data from six rainfall stations (Fig. 1 in the MS) in the basin from 2013 to 2015, and used the annual precipitation contour map data in the Tibet Water Resources Bulletin (2012–2017).

First, the precipitation data from these eight stations were interpolated using the reversed distance squared method. Then, we considered the precipitation–elevation relationship and the topographic effect of the plateau mountainous area, particularly the blocking effect of the mountains on water vapor transport. The watershed was divided into five sub-regions (Fig. c) according to the precipitation contour map, and the vertical precipitation gradient in each sub-region was calculated. Finally, the precipitation was corrected using the elevation and the vertical precipitation gradient to obtain the precipitation data for the entire basin.

The specific methods will be introduced in the supplementary materials in the next MS.

Figure c. Sub-region divisions used for precipitation elevation interpolation.

Since the introduction of snow layer was done in the distributed model setting, we would like to know how the snow cover distribution was estimated, and how snow meltdown was assessed (in spatial terms). Typical snow cover thickness in different parts of the catchment must also be presented.

Reply: Thank you for pointing this out. We apologize for any confusion. We will make it clearer that the precipitation and temperatures of all calculation units are different, and the model can calculate the snowfall amount according to the meteorological data. Based on the mass balance of snow (Equation 16), the amount of snow cover in each calculation unit can be calculated, from which the snow cover distribution in the basin can be obtained.

The basic calculation unit in the model was the contour band. The differences in temperature and precipitation due to altitude difference cause the calculation units at higher altitudes to accumulate more snow and undergo less melting. Units with lower elevations accumulate less snow and undergo more melting. In the model, we established a thickness threshold. When the snow thickness difference between two calculation units exceeded this threshold, snow meltdown occurred. The snow in the higher-altitude calculation unit slides into the next unit until the two units had the same snow thickness.

The temporal and spatial variations in snow thickness are shown in Figure d.